

**Tracing seasonal signals in dry/wet status for regions with**
**simultaneous rain and heat from Eastern and Central Asia**
**since the Last Glacial Maximum**
**Simin Peng[1], Yu Li[1*], Zhansen Zhang[1], Mingjun Gao[1], Xiaowen Chen[1], Junjie**
**Duan[1], Yaxin Xue[1]**
[1]*Key Laboratory of Western China's Environmental Systems (Ministry of Education),*
*College of Earth and Environmental Sciences, Center for Hydrologic Cycle and Water*
*Resources in Arid Region, Lanzhou University, China*
[*]Corresponding author
E-mail address: liyu@lzu.edu.cn (Y. Li).
**Abstract**
The global monsoon region with the summer precipitation regime and the
Mediterranean climate region with the winter precipitation regime showed opposite
dry/wet evolution since the Last Glacial Maximum (LGM). The remarkable difference
in summer precipitation regime and winter precipitation regime reveal the seasonal
signals of precipitation in multi-time scale climate change. Most studies revealed that
the dry/wet status with the summer precipitation regime in Eastern and Central Asia
(EA and CA) contradicted those with the winter precipitation regime in CA. Based on
the comprehensive study of modern observation datasets, model outputs of eight



climate models from the Paleoclimate Model Intercomparison Project phase 3 (PMIP3)
and proxy records from EA and CA, here we show that seasonal signals of precipitation
derived from the simultaneity of rain and heat periods could govern the difference and
linkage in dry/wet status from EA and CA. EOF analysis results of mean annual
precipitation uncover different precipitation regimes in EA and CA. However, the
similarity between EA and the east of CA, indicated by EOF results of summer and
winter precipitation, suggested seasonal signals of precipitation are the primary factor
causing the linkage in dry/wet status at short-term timescales. In particular, summer and
winter precipitation in EA and CA is associated with the Asian monsoon, westerlies,
ENSO, NAO, and PDO. At long-term timescales, the compilation of 42 proxy records
since the LGM in EA and CA reveals parallel dry/wet changes in EA and the east of CA
as well, attributing to seasonal signals triggered by the insolation in different seasons.
PMIP3 multi-model simulation between the LGM and Mid-Holocene (MH) in summer
and winter visually was conducted to analyze paleoclimate mechanisms of difference
and linkage in dry/wet status from EA and CA. Results show that summer insolation
influences the meridional temperature gradient and sea level pressure in the summer,
changing the intensity of the westerly winds and summer monsoon and further
controlling the summer precipitation in EA and the east of CA. Meanwhile, winter
insolation contributes to the general warming in EA and the core region of CA, and in
turn results in lower relative humidity, which ultimately increases winter precipitation
during the LGM. Overall, we suggest, in addition to the traditional difference caused
by different precipitation regimes, that dry/wet status in EA and CA universally have



inter-regional connections affected by seasonal signals of precipitation at multi-time
scales.

**Keywords**
seasonal signals; Eastern Asia (EA); Central Asia (CA); dry/wet status; multi-time
scales; Last Glacial Maximum

**1 Introduction**

As typical midlatitude climatic regions, Eastern and Central Asia (EA and CA) are

commonly featured with vigorous circulations and are dominated by two atmospheric
systems, namely midlatitude westerlies and Asian monsoon (Li, 1990; Zhang and Lin,
1992; Chen et al., 2008; Nagashima et al., 2011). These two regions are generally
characterized by opposite climate and environment changes, embodied in water
resources, vegetation cover and ecosystems, which gives rise to their different response
to climate change (Sorg et al., 2012; Zhang and Feng, 2018). CA, where precipitation
is scarce throughout the year, is the largest arid region in the mid-latitudes dominated
by westerlies (Chen et al., 2009; Huang et al., 2015a). On the contrary, affected by the
Asian summer monsoon that carries water vapor from the Ocean, the monsoon-
dominated EA has more precipitation (Wang et al., 2017). Therefore, exploring
spatiotemporal climate and environment changes in EA and CA has attracted much
research interest.

Over the past few years, there have been many comparative studies for dry/wet





changes at multi-time scales from EA and CA. Early works suggested that the climate
change mode of 'cold-wet' or 'warm-dry' occurred in northwestern China during the
last glacial/interglacial cycle, which is different from the 'cold-dry' or 'warm-wet'
modes of the monsoon climate (Li, 1990; Han and Qu, 1992; Han et al., 1993). Based
on the integration of paleoclimate records, modern meteorological observation data and
paleoclimate simulations, Chen et al. (2008, 2009, 2019) revealed the 'westerlies-
dominated climatic regime' in arid CA from millennium to interdecadal timescales,
which is out-of-phase or anti-phased with the dry/wet status in the monsoon-dominated
regions. However, the paleoclimate records in part regions of CA provided
asynchronous climate evolution history, in contradiction with the dry/wet changes
caused by the westerlies (An et al., 2006; Zhao et al., 2015; Wang et al., 2018). The
latest studies proposed that the persistent weakening of the East Asian summer
monsoon since 1958, causing an increasing contribution of the monsoonal water vapor
transport, thereby enhancing summer precipitation in arid CA (Chen et al., 2021a; Chen
et al., 2021b). Therefore, further research is needed to explain dry/wet changes in
different regions and explore the difference and linkage in climate change modes from
EA and CA at multi-time scales.

The seasonal signals of precipitation derived from the simultaneity of rain and heat

periods, behaving as that the summer half-year at short-term timescales and warm
period at long-term timescales has more precipitation than the winter half-year and cold
period respectively, is an important phenomenon in climate change in EA and CA at the
multi-time scale. This study aims to focus on the transitional zone in the arid and semi-



arid region of eastern CA where the westerlies and the monsoon interact and have the
summer precipitation regime the same as the monsoon-dominated EA. Utilizing
modern observations, paleoclimate proxies, and model simulations, we conducted a
comprehensive analysis for dry/wet status in EA and CA at multi-time scales based on
seasonal signals of precipitation.

**2 Materials and methods**
*2.1 Study area*

CA is the largest arid and semi-arid areas in the mid-latitude hinterland of the

Eurasian continent, extending from the Caspian Sea in the west to the modern Asian
summer monsoon limit in the east, comprising the central Asian countries, NW China,
and southern Mongolian Plateau (Fig. 1). Considering that the strength and trajectory
of monsoon circulation is a major control on moisture in EA, we viewed the monsoon
China in the east and south of the modern Asian summer monsoon limit as EA (Fig. 1).
We calculated the precipitation difference between the summer (April, May, June, July,
August, and September) and winter (January, February, March, October, November,
and December) half year over 1971-2020, and then defined the region greater than 0
mm as the simultaneous region of rain and heat periods (Fig. 1, gray slash). Eastern CA
belongs to the simultaneous region of rain and heat periods. The seasonality perspective
implies that different precipitation regimes could affect the difference and linkage in
climate change modes from EA and CA at the multi-time scale. Taking seasonal signals
as the dividing criteria, the core region of CA is characterized by a wet cold-season



climate, whereas EA and eastern CA is characterized by a wet warm-season climate
(Fig. 1).

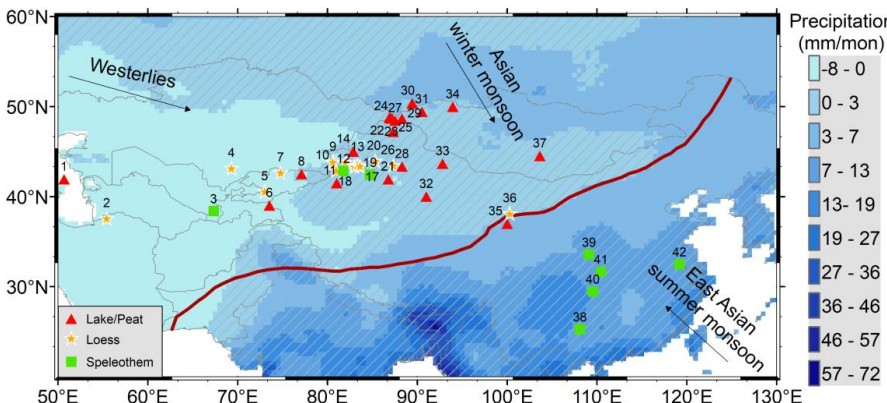


**Figure. 1** Overview map showing the paleoclimate record sites selected in this study from EA and CA, the difference
between summer and winter precipitation over 1965-2014 (shade), and the dominant circulation systems, including
the westerlies, Asian winter monsoon and East Asian summer monsoon. The modern Asian summer monsoon limit
(red solid line) is summarized by Chen et al. (2008, 2019). The gray slash represents the simultaneous region of the
rain and heat periods.

***2.2 Modern observation and analytical methods***

The monthly high-resolution (0.5°×0.5°) land precipitation data (referred to as

CRU TS4.06) are selected from a Climatic Research Unit (CRU) updated gridded
climate dataset in the University of East Anglia (van der Schrier et al., 2013; Harris et
al., 2014; Barichivich et al., 2021). The CRU monthly climate archives obtain from the
auspices of the World Meteorological Organization (WMO) in league with the US
National Oceanographic and Atmospheric Administration (NOAA, via its National
Climatic Data Center, NCDC). Global Reanalysis 1 dataset including monthly mean
geopotential height, zonal wind, and meridional wind is collected from the National
Centers for Environmental Prediction/National Center for Atmospheric Research


(NCEP/NCAR) (Kalnay et al., 1996). The reanalysis datasets have a horizontal
resolution of 2.5° in latitude and longitude and a vertical resolution of 17 pressure levels
from 1000 to 10 hPa. The high-resolution monthly averaged data high resolution for
the vertical integral water vapor from the European Centre for Medium-Range Weather
Forecasts (ECMWF) reanalysis v5 (ERA5), intending to be used as a meteorological
forcing dataset for land surface and hydrological models, is used in this study. This
dataset is from 1979 to the present with a spatial resolution of 0.25° in latitude and
longitude and a single level integrated from the surface to the top of the atmosphere
(Hersbach et al., 2020).
We used the National Centers for Environmental Information (NCEI) Pacific
Decadal Oscillation (PDO) index based on NOAA's extended reconstruction of SSTs
(ERSST Version 5) to analyze long-lived El Niño-like pattern of Pacific climate
variability (Zhang et al. 1997; Mantua and Hare, 2002). The data can be obtained at
https://www.ncei.noaa.gov/pub/data/cmb/ersst/v5/index/ersst.v5.pdo.dat. The Niño 3.4
index is the most commonly used index to define El Niño and La Niña events. We
selected the Niño 3.4 of area-averaged SST from 5°S-5°N and 170-120°W using the
HadISST1   dataset   (Rayner   et   al.,   2003).   The   data   can   be   obtained   at
https://psl.noaa.gov/gcos_wgsp/Timeseries/Nino34/. Positive values of the North
Atlantic Oscillation (NAO) index are typically associated with stronger midlatitude
westerlies and increased water vapor content from the North Atlantic. We used the
Hurrell NAO index (station-based) to investigate the impact factor of midlatitude
westerlies (Hurrell, 1995; Hurrell and Deser, 2009). The data can be obtained at



https://climatedataguide.ucar.edu/sites/default/files/2022-10/nao_station_monthly.txt.
Empirical orthogonal function (EOF) is a powerful method for dimensionality
reduction and pattern extraction. EOF can decompose multidimensional climate data
from different locations into spatial (EOF modes) and temporal functions (principal
components). Therefore, to investigate the spatiotemporal variations of precipitation at
the interannual timescale over EA and CA, the EOF analysis was applied to the CRU
TS4.06 gridded precipitation data and ERA5 vertical integral water vapor. We focused
on the first two leading modes that objectively account for the majority of dry/wet status
in EA and CA (Lorenz, 1956).

***2.3 Calculation of Monsoon and westerly wind index***
The East Asian summer monsoon index (EASMI) is defined as the 850 hPa
average summer meridional wind speed from June to August over (27°N~37°N,
110°E~120°E) encompassing the East Asian summer monsoon domain (Liu et al.,
2014). The equation is as follows:
$$\text{EASMI} = \overrightarrow{V_{850}}(27°\text{~}37°\text{N}, 110°\text{~}120°\text{E})$$
The westerly wind index (WWI) is defined as the zonal difference of the 500 hPa
averaged geopotential height over (35°N~50°N, 70°E~110°E) (Li et al., 2008). The
equation is as follows:
$$\text{WWI} = \overline{H_{35°}} - \overline{H_{50°}} = \frac{1}{17}\left[\sum_{\gamma=1}^{17} H(\gamma, 35°\text{N}) - \sum_{\gamma=1}^{17} H(\gamma, 50°\text{N})\right]$$
where $H$ is the 500 hPa average height geopotential, $\gamma$ is the number of longitudes
taken along the latitude circle with a spacing of 2.5°.



The East Asian winter monsoon index (EAWMI) is defined as the difference
between the 300 hPa averaged zonal wind speed from December to February over
(27.5°~37.5°N, 110°~170°E) and (50°~60°N, 80°~140°E) (Jhun and Lee, 2004). The
equation is as follows:
EAWMI = $\overrightarrow{U_{300}}$(27.5°~37.5°N, 110°~170°E)-$\overrightarrow{U_{300}}$(50°~60°N, 80°~140°E)
The calculation of EASMI, WWI, and EAWMI all rely on the NCEP Reanalysis 1
dataset.

*2.4 Regional paleoclimatic proxy data*
Here we compiled various paleoclimate records to reconstruct long-term climate
variability and primarily paid close attention to paleo-precipitation and moisture
changes since the LGM. We set three criteria to collect all the published proxy records
from EA and CA in our study. Firstly, the records should be located primarily in the
intersection encompassing the simultaneous region of rain and heat periods in EA and
CA, which is in favor of investigating the difference and linkage in climate change
modes from EA and CA. Accordingly, some typical records climatologically influenced
by midlatitude westerlies in cores of EA and CA were selected for comparative analysis.
Secondly, the proxies should be clearly indicative of changes in effective moisture or
precipitation which have been confirmed by the original investigators. Third, the record
length should cover the most period since LGM without documented depositional
hiatuses. Fourth, the fluctuation and variation of proxy records should be predominantly
forced by climate change, rather than human activities (Manoj et al., 2020; Chen et al.,



2021c, 2022). Following the above criteria, a total of 42 proxy records from lakes, peats,
loess, and stalagmites since the LGM were compiled for EA and CA (Fig. 1), enabling
us to comprehensively review the LGM moisture evolution of the region. In light of
seasonal signals of precipitation, 35 records are from the summer precipitation region,
and seven records are from the winter precipitation region. Detailed information about
these selected proxy records is presented in Table 1.
**Table 1.** Paleoclimate records selected in this study.

| Code | Section name | Record type | Lat | Lon | Evaluation (m a.s.l) | Precipitation regime | Dating materials | Dating Method | Time period (cal ka BP) | Proxy | Proxy indication | References |
|---|---|---|---|---|---|---|---|---|---|---|---|---|
| 1 | Caspian Sea | Lake | 41.93 | 50.67 | -28 | winter | Ostracods | $^{14}$C | 12.4-2.4 | Pollen | Moisture | Leroy et al. (2014) |
| 2 | YE section | Loess | 37.60 | 55.43 | 383 | winter | Quartz | OSL | 11.8-0 | $\delta^{13}C_{org}$ | Moisture | Wang et al. (2020) |
| 3 | Ton Cave | Speleothem | 38.40 | 67.34 | 3226 | winter | Carbonate | U-Th | 135-0 | $\delta^{13}C$ | Moisture | Cheng et al. (2016) |
| 4 | Valikhanov section | Loess | 43.17 | 69.31 | 1000 | winter | Bulk organic matter, charcoal | $^{14}$C | 46-0 | $\delta^{13}C$ | Moisture | Ran and Feng (2014) |
| 5 | Osh section | Loess | 40.61 | 73.01 | 1038 | winter | Humin | $^{14}$C | 30-0 | Grain-size, MS | Effective moisture | Li et al. (2021) |
| 6 | Lake Karakul | Lake | 39.02 | 73.53 | 3915 | winter | plant remains, bulk sediments, living charophyte | $^{14}$C | ~29-0 | TIC, TOC, C/N, Grain-size, $\delta^{13}C_{carb}$, $\delta^{18}O$ | Moisture | Heinecke et al. (2017) |
| 7 | BSK section | Loess | 42.70 | 74.78 | 1432 | winter | Bulk organic matter | $^{14}$C | 26-0 | Grain-size, MS, color proxies | Moisture | Li et al. (2020a) |
| 8 | Lake Issyk-Kul | Lake | 42.50 | 77.10 | 1607 | summer | Bulk sediments | $^{14}$C | 12.75-3.6 | $\delta^{18}O$, $\delta^{13}C$, Pollen, CaCO3, MS | Moisture | Ricketts et al. (2001); Leroy et al. (2021) |
| 9 | HC14 section | Loess | 43.88 | 80.60 | 554 | summer | Bulk organic matter | $^{14}$C | 10-0 | MS | Moisture | Jia et al. (2021) |
| 10 | ZS section | Loess | 42.93 | 80.96 | 1650 | summer | Quartz | OSL | 12.6-0 | Grain-size, MS | Moisture | Kang et al. (202 |





| | | | | | | | | | | | | |
|---|---|---|---|---|---|---|---|---|---|---|---|---|
| 11 | Lake Sayram | Lake | 41.50 | 81.03 | 2072 | summer | Bulk sediments | $^{14}$C | 13.8-0 | Pollen | Moisture | 0) Jiang et al. (2013, 2022) |
| 12 | Kesang Cave | Speleothem | 42.87 | 81.75 | ~2000 | summer | Carbonate | U-Th | 22.8-0 | $\delta^{18}$O | Precipitation | Cheng et al. (2012, 2016) |
| 13 | Yili section | Loess | 43.86 | 81.97 | 928 | summer | Charcoal | $^{14}$C | 15-0 | A/C ratio | Moisture | Li et al. (2011) |
| 14 | Lake Aibi | Lake | 45.01 | 82.86 | 200 | summer | Bulk sediments | $^{14}$C | 13.8-0 | Pollen | Moisture | Wang et al. (2013) |
| 15 | XEB section | Loess | 43.42 | 82.99 | 888 | summer | Quartz | OSL | 12-0 | Grain-size, MS | Moisture | Kang et al. (2020) |
| 16 | TLD16 section | Loess | 43.36 | 83.02 | 1567 | summer | Quartz | OSL | 20-0 | MS | Moisture | Jia et al. (2021) |
| 17 | ZKT section | Loess | 43.53 | 83.30 | 846 | summer | Bulk organic matter | $^{14}$C | 16-0 | MS | Moisture | Chen et al. (2016); Jia et al. (2021) |
| 18 | KS16 section | Loess | 43.43 | 83.62 | 1314 | summer | Quartz | OSL | 12-0 | MS | Moisture | Jia et al. (2021) |
| 19 | Baluk Cave | Speleothem | 42.43 | 84.73 | 2400 | summer | Carbonate | U-Th | 9.3-0 | Trace elements | Moisture | Liu et al. (2020) |
| 20 | LJW10 section | Loess | 43.97 | 85.33 | 1462 | summer | Quartz and K-feldspar | OSL | 16-0 | MS | Moisture | Chen et al. (2016) |
| 21 | Lake Bosten | Lake | 41.94 | 86.76 | 1048 | summer | Bulk organic matter, plant, tree leaves | $^{14}$C | 8.2-0 | Pollen | Moisture | Huang et al. (2009) |
| 22 | Narenxia peat | Peat | 48.80 | 86.90 | 1760 | summer | Bulk peat, lake mud | $^{14}$C | 11.8-0 | Pollen, $\delta^{13}$C | Annual precipitation | Feng et al. (2017); Zhang and Feng (2018) |
| 23 | Lake Kanas | Lake | 48.70 | 87.01 | 1365 | summer | Terrestrial plant macrofossils | $^{14}$C | 13.4-0 | Pollen | Annual precipitation | Huang et al. (2018) |
| 24 | Big Black peat | Peat | 48.68 | 87.18 | 2168 | summer | Cellulose | $^{14}$C | 9.5-0 | Pollen, $\delta^{18}$O, $\delta^{13}$C | Moisture | Xu et al. (201 |



| | | | | | | | | | | | | |
|---|---|---|---|---|---|---|---|---|---|---|---|---|
| | | | | | | | | | | | | 9) |
| 25 | Lake Wulungu | Lake | 47.20 | 87.29 | 479 | summer | Bulk organic matter | [14]C | 9.5-0 | Pollen, $\delta^{13}$C, grain-size | Moisture | Liu et al. (2008) |
| 26 | ZL section | Loess | 43.50 | 87.33 | 1756 | summer | K-feldspar | OSL | 10.8-0 | MS | Moisture | Chen et al. (2016); Gao et al. (2019) |
| 27 | Tuole haite peat | Peat | 48.44 | 87.54 | 1700 | summer | Plant residuals | [14]C | 10.6-0 | Pollen | Moisture | Zhang et al. (2020) |
| 28 | Chaiwopu peat | Peat | 43.35 | 88.30 | 800 | summer | Plant, Bulk sediments | [14]C | 11.5-0 | Pollen | Moisture | Yang et al. (2021) |
| 29 | Hoton Nurr | Lake | 48.67 | 88.30 | 2083 | summer | Bulk sediments | [14]C | 11.5-0 | Pollen | Annual precipitation | Rudaya et al. (2009) |
| 30 | Lake Akkol | Lake | 50.38 | 89.42 | 2204 | summer | Bulk sediments | [14]C | 10-0 | Pollen | Vegetation change | Blyakharchuk et al. (2007) |
| 31 | Achit Nuur | Lake | 49.42 | 90.52 | 1444 | summer | Bulk sediments, root, mollusk | [14]C | 22.6-0 | $\delta^{18}$O | Annual precipitation | Sun et al. (2013) |
| 32 | Lake Lup-Nur | Lake | 40.00 | 91.00 | 780 | summer | Quartz | OSL | 9-0 | Soluble salt content, grain-size, pollen, ostracod | Moisture | Liu et al. (2016) |
| 33 | Lake Balikun | Lake | 43.67 | 92.80 | 1575 | summer | Bulk organic matter, plant macrofossils, pollen; | [14]C | 29.1-0 | Pollen | Moisture | Tao et al. (2010); An et al. (2012); Zhao et al. (2015) |
| 34 | Bayan Nurr | Lake | 49.98 | 93.95 | 932 | summer | Bulk sediments | [14]C | 15-0 | Pollen | Annual precipitation | Tian et al. (2014) |
| 35 | Qinghai Lake | Lake | 37.00 | 100.00 | 3200 | summer | Bulk organic matter | [14]C | 18-0 | $\delta^{18}$O | summer monsoon precipitation | Shen et al. (2005); Liu et al. (2007) |
| 36 | Qilian section | Loess | 38.16 | 100.27 | 2810 | summer | Bulk organic matter | [14]C | 22-0 | $\delta^{18}$O, $\delta^{13}$C | Effective moisture | Li et al. (2020b) |
| 37 | Lake Ulaan | Lake | 44.53 | 103.63 | 1024 | summer | Bull samples, quartz | [14]C, OSL | 17-0 | TOC | Moisture | Lee et al. (2011, 201 |





| | | | | | | | | | | | | 3) |
|---|---|---|---|---|---|---|---|---|---|---|---|---|
| 3 8 | Dong ge Cave | Spe leot he m | 25 .2 8 | 10 8. 08 | 680 | summ er | Carbonate | U-Th | 16-0 | δ18O | summer monsoon precipitati on | Dyk oski et al. (200 5) |
| 3 9 | Jiuxia n Cave | Spe leot he m | 33 .5 66 7 | 10 9. 1 | 1495 | summ er | Carbonate | U-Th | 19-0 | δ18O | summer monsoon precipitati on | Cai et al. (201 0) |
| 4 0 | Lianh ua Cave | Spe leot he m | 29 .4 83 | 10 9. 53 3 | 455 | summ er | Carbonate | U-Th | 12.5-0 | δ18O | summer monsoon strength | Zha ng et al. (201 3) |
| 4 1 | Sanba o Cave | Spe leot he m | 31 .6 67 | 11 0. 43 3 | 1900 | summ er | Carbonate | U-Th | 13-0 | δ18O | Summer rainfall | Don g et al. (200 9) |
| 4 2 | Hulu Cave | Spe leot he m | 32 .5 0 | 11 9. 17 | 90 | summ er | Carbonate | U-Th | Nov-75 | δ18O | summer monsoon precipitati on | Wan g et al. (200 1) |






*2.5 Paleoclimatic simulations*

The Paleoclimate Modeling Intercomparison Project (PMIP) was launched to

coordinate and encourage the systematic study of General Circulation Models (GCMs)
and to understand the mechanisms of climate change and the role of climate feedback
(Joussaume et al., 1999) (Table 2). Eight coupled GCMs covering the LGM or MH from
the PMIP3 database were selected to analyze the mechanisms of climate change in this
study (Table 3), including bcc-csm1-1, CNRM-CM5, CCSM4, CSIRO-Mk3-6-0,
GISS-E2-R, MIROC-ESM, FGOALS-s2, and MRI-CGCM3. The output data of the
PMIP3 in the LGM and MH are available at htQTPs://esgf-node.llnl.gov/search/esgf-
llnl/. By chiefly interpolating various climate variables on the common 1°×1° grid and
then sorting the values of model simulations from minimum to maximum, we extracted
the median value of all PMIP3 models used in this paper to evaluate the PMIP3 model
simulations and acquire the scientific model simulation value.
**Table 2.** Boundary conditions and forcing for PMIP3-CMIP5 models at the LGM and MH.

| Period | Eccentricity | Obliquity (°) | Longitude of perihelion (°) | $CO_2$ (ppm) | $CH_4$ (ppb) | $N_2O$ (ppb) | Ice sheet | Vegetation |
|--------|--------------|---------------|------------------------------|--------------|--------------|--------------|-----------|------------|
| LGM | 0.018994 | 22.949 | 114.425 | 185 | 350 | 200 | Peltier (2004), 21 ka | Present day |
| MH | 0.018682 | 24.105 | 0.87 | 280 | 650 | 270 | Peltier (2004), 0 ka | Present day |


**Table 3.** Basic information about climate models from PMIP3-CMIP5 used in this study.

| Model | Institute | Resolutions | Variables[*] | References |
|-------|-----------|-------------|-----------|------------|
| bcc-csm1-1 | Beijing Climate Center, China Meteorological Administration, China | 64×128 (17) | ua, va, zg, hus, psl, pr, tas | Randall et al. (2007) |
| CNRM-CM5 | Centre National de Recherches Météorologiques, France | 128×256 (17) | ua, va, zg, hus, psl, pr, tas | Voldoire et al. (2013) |





| CCSM4 | National Center for Atmospheric Research, USA | 288×192 (17) | ua, va, zg, hus, psl, pr, tas | Gent et al. (2011) |
|---|---|---|---|---|
| CSIRO-Mk3-6-0 | Australian Commonwealth Scientific and Industrial Research Organization Marine and Atmospheric Research in collaboration with the Queensland Climate Change Centre of Excellence, Australia | 96×192 (18) | ua, va, zg, hus, psl, pr, tas | Rotstayn et al. (2010) |
| GISS-E2-R | NASA Goddard Institute for Space Studies, USA | 144×90 (17) | ua, va, zg, hus, psl, pr, tas | Schmidt et al. (2014) |
| MIROC-ESM | Japan Agency for Marine-Earth Science and Technology, Japan | 128×64 (35) | ua, va, zg, hus, psl, pr, tas | Watanabe et al. (2011) |
| FGOALS-s2 | LASG-CEES. China | 108×128 (17) | ua, va, zg, hus, psl, pr, tas | Briegleb et a1. (2004) |
| MRI-CGCM3 | Meteorological Research Institute, Japan | 320×160 (23) | ua, va, zg, hus, psl, pr, tas | Yukimoto et al. (2012) |

*: ua means eastward_wind; va means northward wind; zg geopotential Height; hus near-surface relative humidity; psl means sea surface
pressure; pr means precipitation; tas means near-surface temperature

## 3. Results

### *3.1 Seasonal signals at short-term timescales*

To obtain the spatial distribution characteristics of the precipitation anomalies in
EA and CA under the context of seasonal signals, we conducted an EOF analysis on the
precipitation standardized anomaly field over 1971-2020. Figure 2a-d shows the spatial
distribution and time series of EOF decomposition of mean annual precipitation. The
variance contribution rate of the first mode is 10.29%, showing an obvious dipole mode.
The center of negative values is in the core region of CA mainly belonging to the winter
precipitation regime, while the positive values are in the south and north of EA located
in summer precipitation regions (Fig. 2a). This opposite distribution indicates that the
mean annual precipitation in EA and CA have a see-saw pattern. Additionally, the first

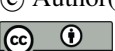

mode exhibits interdecadal and interannual changes according to the PC1 (Fig. 2b). The
variance contribution rate of the second mode is 8.79%, indicating zonal dipole
distribution characteristics (Fig. 2c). The center of positive values is in the north of EA,
and the center of negative values is in the north of CA, also displaying the spatial
diversity of mean annual precipitation in EA and CA (Fig. 2c).

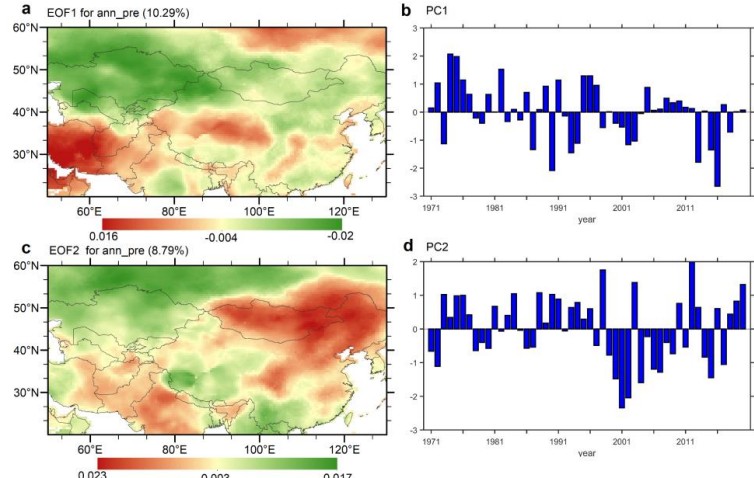


**Figure. 2** The EOF modes and corresponding time series of annual mean precipitation in EA and CA over 1971-
2020.

In order to further explore the contribution of seasonal signals of precipitation to

dry/wet status in EA and CA, we conducted the EOF analysis on the seasonal
precipitation in spring, summer, autumn, and winter. The variance contribution rate of
the first mode in precipitation of four seasons is shown in Fig. 3. The first mode of
spring and autumn precipitation does not show obvious distribution characteristics, and
the contribution rate is relatively uniform, indicating that spring and autumn
precipitation have no special precipitation contribution to EA and CA (Fig. a and c). In
summer precipitation, the centers of positive values are mainly distributed in the north
of EA, while the negative values are mainly distributed in CA and south of EA (Fig.



3b). This spatial distribution indicates that summer precipitation mainly affects the
dry/wet status in northern EA and the east of CA belonging to the simultaneous region
of rain and heat periods, which is in contrast to the core region of CA. In winter
precipitation, the center of the positive value is located in the CA and north of EA,
showing the significant contribution of winter precipitation to CA (Fig. 3d). It is worth
noting that a certain degree of similarities exists in both summer and winter
precipitation of EA and CA, indicating the impact of seasonal precipitation on the
linkage of dry/wet status in EA and CA at short-term timescales.

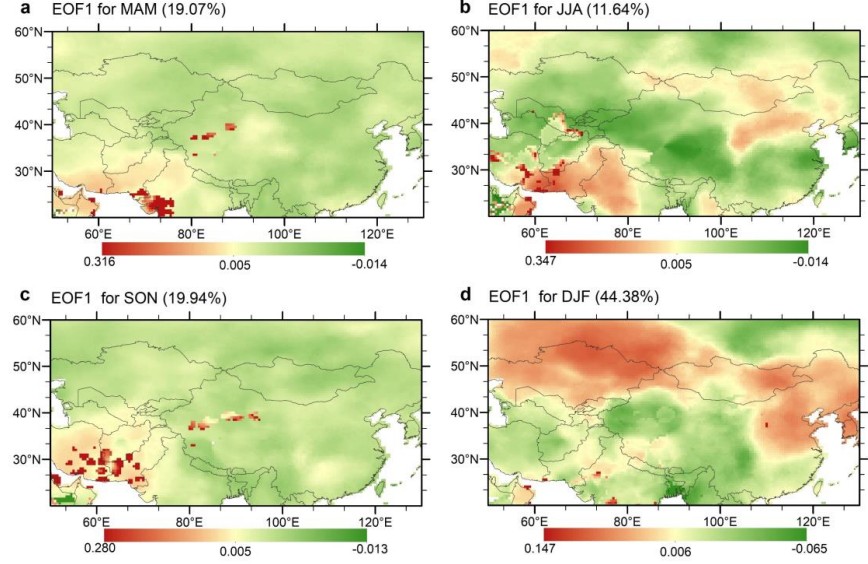


**Figure. 3** The first EOF modes of precipitation in spring (March, April, and May, MAM) (a), summer (June, July,
and August, JJA) (b), autumn (September, October, and November, SON) (c), and winter (December, January, and
February, DJF) (d) in EA and CA over 1971-2020.

Existing studies emphasized the role of water vapor sources in affecting interannual

to interdecadal variability of precipitation (Chen and Huang, 2012; Huang et al., 2015a;
Peng and Zhou, 2017; Wei et al., 2017). Therefore, by analyzing the EOF results of
water vapor content in the whole layer, this study investigates the general characteristics





267 of the spatial distribution of water vapor in EA and CA and discusses the influence

268 mechanism of seasonal signals on dry/wet status in EA and CA at short-term timescales.

269 The EOF1 of the mean annual water vapor shows that the core region of CA is

270 dominated by positive values, while EA and eastern CA are synchronized with negative

271 values (Fig. 4a). The same spatial distribution mode is also reflected in the EOF1 of

272 water vapor difference between summer and winter half-year. To summarize, the water

273 vapor in EA and CA shows a dipole out-of-phase pattern between the simultaneous

274 region of rain and heat periods and the non-simultaneous region of rain and heat periods

275 (Fig. 4b). This implies that the content and source of water vapor are the important

276 reason why the dry/wet status in eastern CA is linked to that in EA by seasonal signals

277 of precipitation.



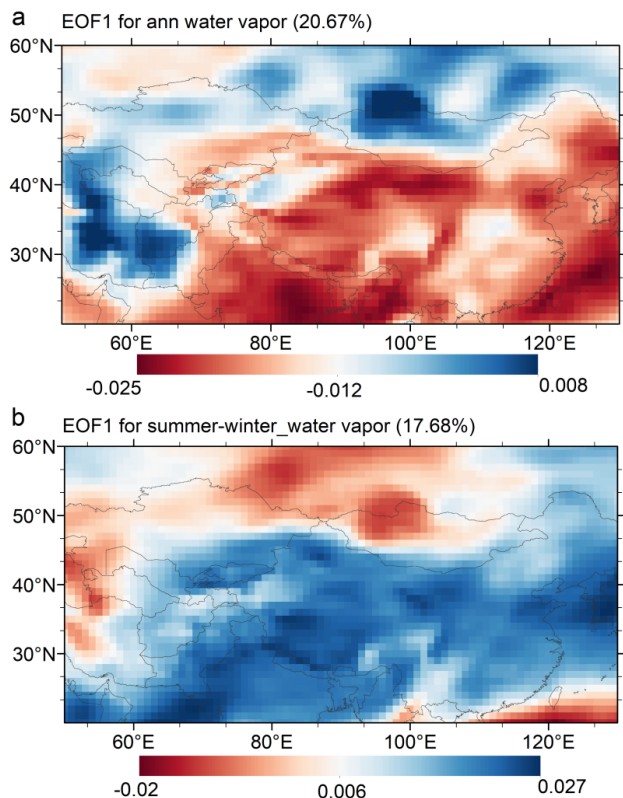

**Figure. 4** a, the EOF1 modes of annual mean integral water vapor in EA and CA over 1979-2018; b, the EOF1 modes
of integral water vapor difference between summer and winter in EA and CA over 1979-2018.

*3.2 Spatiotemporal variation of dry/wet status and seasonal signals at long-term*
*timescales*
In the last decade, many paleoclimate records with a relatively high resolution,
reliable chronology, and unambiguous proxies have been published to discuss the long-
term timescale climate evolution in EA and CA. Forty-two moisture records from
individual sites are used to illustrate the spatiotemporal pattern of dry/wet status since
the LGM in EA and CA (Fig. 7). During the LGM, most regions in EA and CA are in
moderately dry condition (Fig. 7a). However, moderately wet and wet conditions partly
exist in the east of CA. According to the model simulation, Yu et al. (2000) concluded



that the low temperature in the cold period causes decreasing evaporation, with the
enhanced westerlies driven by expanding land ice sheets, forming the high lake level in
western China and the low lake level in eastern China during the LGM. During the early
Holocene (EH), CA is dominated by a dry climate, while EA is moderately wet. At the
same time, there were many records in the east of CA similar to the dry/wet status of
EA. During the MH, the dry/wet status is mainly wet in the core region of CA and
gradually turns into moderately wet and even dry conditions in the east of CA, while
the EA remains moderately wet. By the late Holocene (LH), the EA is characterized by
dry status, while CA is wet. In particular, the dry condition during the LGM and the wet
climate during the EH and MH also reflect another meaning of seasonal signals derived
from the simultaneity of rain and heat periods at long-term timescales, namely the "dry-
cold" pattern and "wet-warm" pattern.

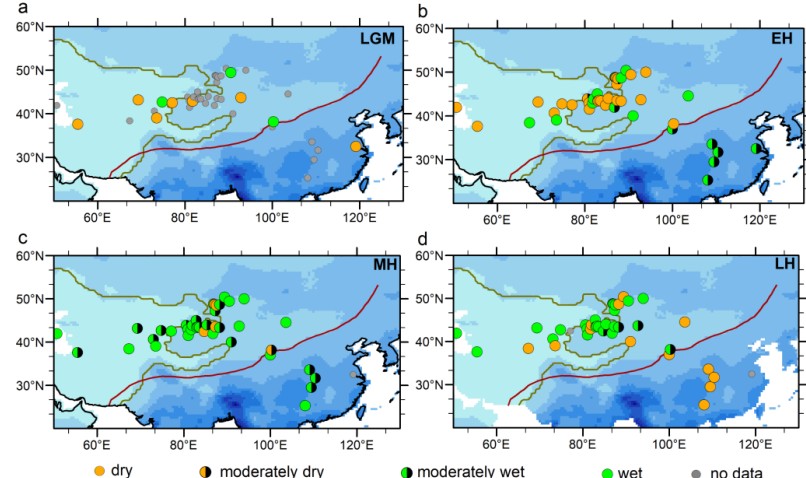


**Figure. 7** Spatio-temporal characteristics of the dry/wet status from 42 records since the LGM, based on the
confirmation of original investigators during the LGM, early Holocene (EH), mid Holocene (MH), and late Holocene
(LH). Records with an incomplete stage are shown by a gray dot. Four summarized levels of dry/wet status: wet,
moderately wet, moderately dry, and dry.

In detail, we further performed a comparative analysis of time series of typically





proxy record in EA and CA (Fig. 8). The reconstructed precipitation covering the past
22,600 years from Achit Nuur suggests the wet periods from 22,600 to 13,200 cal BP
(Fig. 8c). Pollen record from the Caspian Sea, controlled by the westerlies, displays that
the terrestrial vegetation around the Caspian Sea changed from desert/desert steppe
during the last glacial to dry shrubland/forest during the Holocene, revealing the
continuous wetting process since the EH and the wettest LH (Fig. 8a). Meanwhile,
results of climatically-sensitive magnetic properties from the Xinjiang loess
demonstrate that the relatively wet conditions are generally formed after ~6,000 cal BP,
with the wettest climate occurring during the LH (Fig. 8b). However, there is still
partially contradictory for dry/wet changes on long-term timescales in CA, which are
different from CA but similar to EA. Herzschuh. (2006) comprehensively analyzed 75
paleoclimatic records in CA and revealed that wet conditions occurred during the EH
and MH, while the LGM was characterized by the dry climate (Fig. 8h), indicating the
similarity with the monsoon climate represented by the speleothem $\delta^{18}O$ records from
Dongge Cave and Hulu Cave (Fig. 8d). High precipitation in the EH and MH, indicated
by $\delta^{18}O$ records of ostracod shells from Qinghai Lake, shows that the climate in Qinghai
Lake since the late glacial reflects the monsoon-dominated characteristic (Fig. 8e). The
climate in Ulaan Nuur is wettest during the EH, humid during the MH and dry in the
LH, embodying a typical characteristic of the East Asian summer monsoon (Fig. 8f).
Based on the sediment cores from Lake Karakul and Lake Issyk-Kul, the EH and MH
is characterized by wetter conditions in the region, and the lake level remained low
during the LGM (Fig. 8g and j). Furthermore, the regional climate in western China,



inferred from the speleothem oxygen-carbon isotope in Kesang Cave, suggests a close
coupling with the Asian summer monsoon (Fig. 8i). The lake level and climate
reconstructed results also conducted that the "dry-cold" pattern triggered a substantial
lowering of lake level in most of arid western China, challenging the traditional view
of "wet-cold" pattern and high lake levels during the LGM (Zhao et al., 2015).

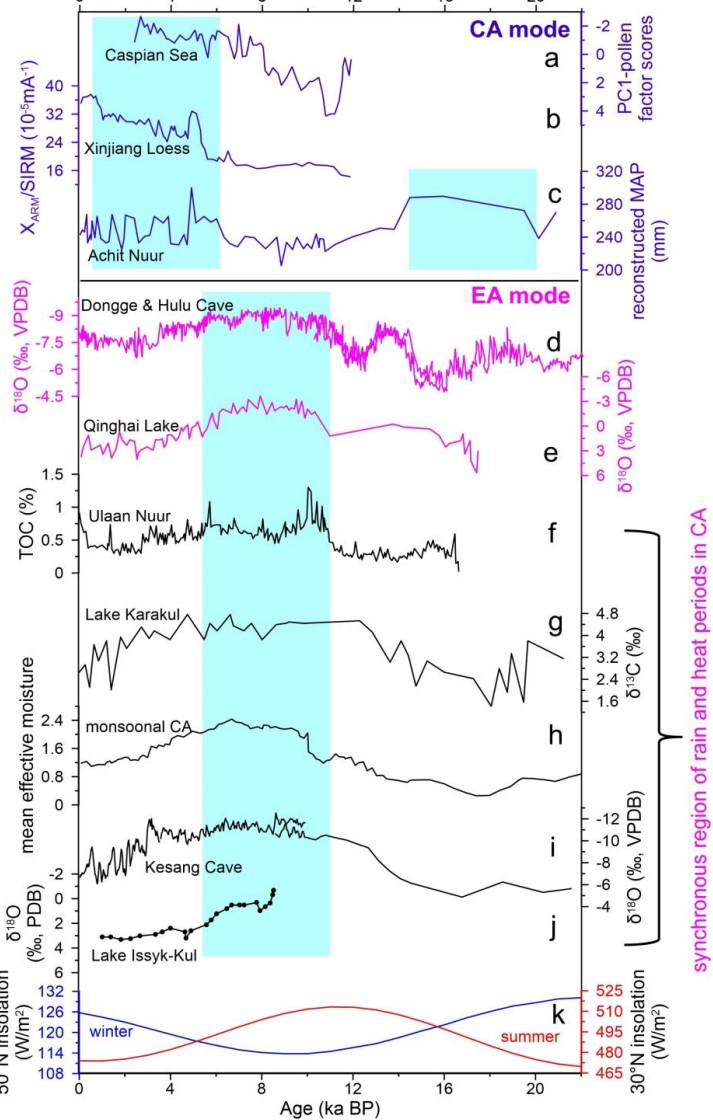






**Figure. 8** A comparison of proxy variability recorded in EA and CA. a Pollen record from the Caspian Sea (Leroy
et al., 2014); b X$_{ARM}$/SIRM in the LJW10 section of the Xinjiang Loess (Chen et al., 2016); c Reconstructed MAP
(mean annual precipitation) from Achit Nuur (Sun et al., 2013); d speleothem δ$^{18}$O values records from Dongge
Cave and Hulu Cave (Yuan et al., 2004; Wang et al., 2001); e δ$^{18}$O of ostracode shells from Qinghai Lake (Liu et al.,
2007); f TOC (Total organic carbon) from Ulaan Nuur (Lee et al., 2013); g δ$^{13}$C from Lake Karakul (Heinecke et al.,
2017; Mischke et al., 2017); h Mean effective moisture from monsoonal Central Asia (Herzschuh, 2006); i δ$^{18}$O from
Kesang Cave (Cheng et al., 2016); j δ$^{18}$O from Lake Issyk-Kul (Ricketts et al., 2001); k Summer (red line) insolation
at 30°N and winter (blue line) insolation at 50°N (Berger, 1978);. Blue shadows indicate the wet period of
paleoclimate proxies.

## 4. Discussion

### 4.1 Possible dynamics of seasonal signals at short-term timescales

EOF analysis of precipitation and water vapor consistently verifies that the
connection between EA and the east of CA exists under the traditional differentiation
between EA and the core region of CA. Considering that the east of CA is present as
the summer precipitation regime. Therefore, we propose that seasonal signals of
precipitation contribute to the connection between EA and the east of CA.



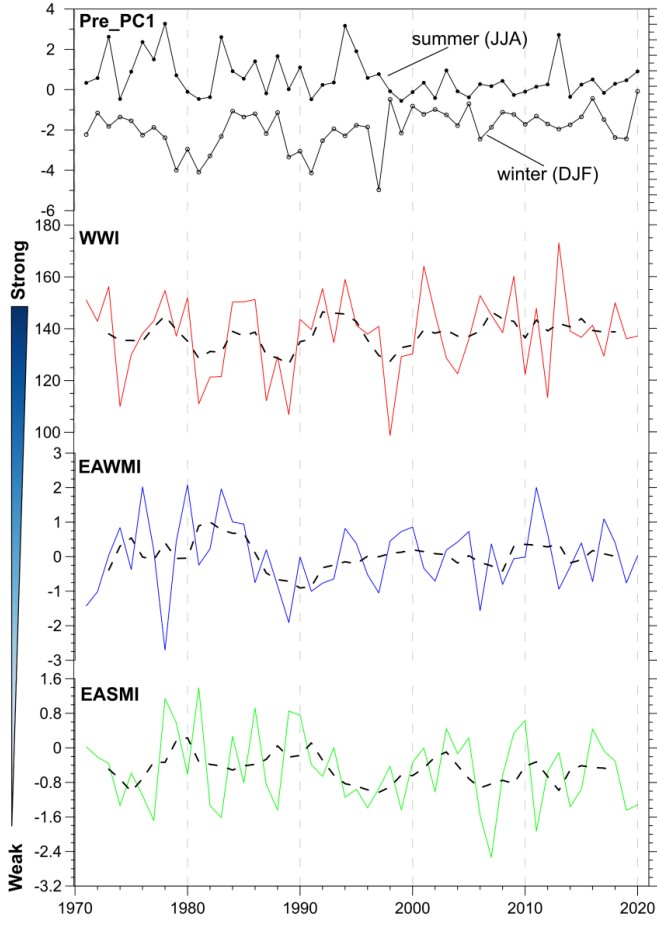

**Figure. 5** The time series of the precipitation PC1 in summer, winter, WWI, EAWMI, and EASMI over 1971-2020.

Generally, atmosphere circulations have important effects on the spatial distribution and the transportation of water vapor. In order to explore the influence of the modern air-sea circulation system on the summer and winter precipitation, we analyzed the time series of the precipitation PC1, WWI, EAWMI, EASMI, NAO, PDO, and ENSO over 1971 to 2020 (Fig. 5 and 6). Comparing the winter precipitation PC1 with WWI and EAWMI (Fig. 5), the weakening of the westerlies and winter monsoons is usually accompanied by an increase in winter precipitation. However, there is not a





significant relationship between PC1 of summer precipitation and EASMI. As shown
in Figure. 6, summer PDO and ENSO are basically similar to winter PDO and ENSO.
However, the markable discrepancy exists in the evolution of winter NAO and summer
NAO. The NAO and ENSO index presents interannual timescale variation, and the
PDO index has an interdecadal timescale cycle. The NAO index and the winter
precipitation PC1 have a positive correlation, suggesting that the North Atlantic may
have certain effects on the winter precipitation through the air-sea interaction. Positive
values of the NAO index are usually accompanied by stronger midlatitude westerlies
and increased water vapor content from the North Atlantic. The PDO and ENSO,
however, were related to the summer precipitation PC1. The development of winter
precipitation at interdecadal timescales was not connected with PDO, whereas there is
a positive correlation before the 2000s between summer precipitation and PDO.



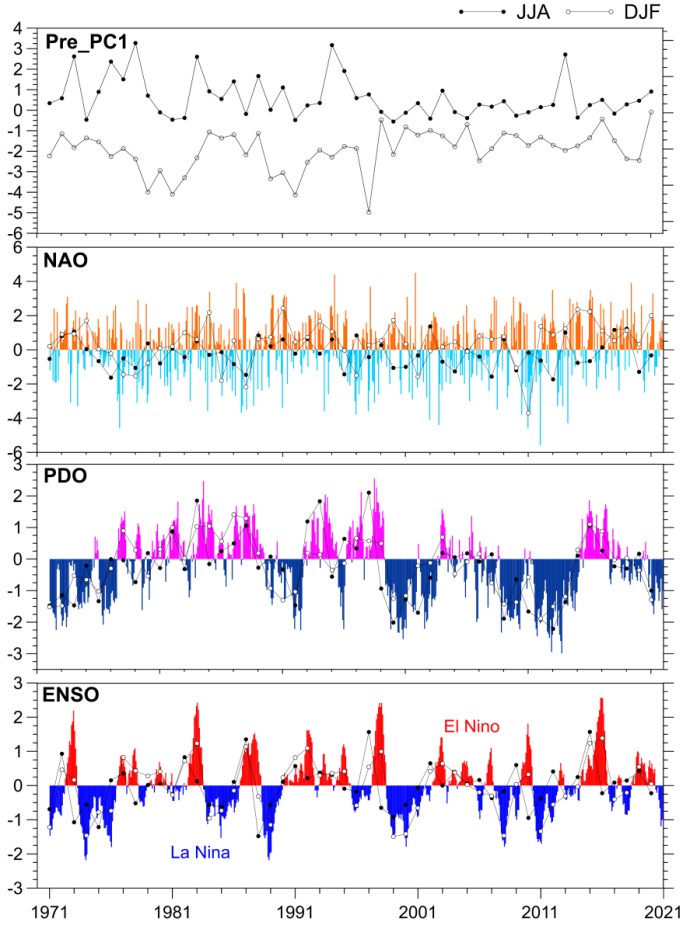


**Figure. 6** The time series of the precipitation PC1 in summer, winter, and annual mean, NAO, PDO, and
ENSO over 1971-2020.

A majority of relevant studies stand for that precipitation variations in CA are
subjected to water vapor transported by the mid-latitude westerlies, where the
monsoonal water vapor source is hard to reach (Huang et al., 2015a; Guan et al., 2019).
Abundant moisture is brought to CA from polar airmass, North Atlantic and the eastern
Mediterranean Sea, and continues to diffuse eastward to the arid region of northwest
China (Lioubimtseva, 2014). Meanwhile, several studies in recent years found that the
anti-phase pattern between the East Asian summer monsoon and the westerlies causes



the seesaw phenomenon of precipitation variation in northwest China (the east of CA
in this study) (Zhang et al., 2019; Wu et al., 2019). However, Chen et al. (2021a)
proposed that the East Asian summer monsoon plays an important role in the
interdecadal variability of summer precipitation in CA through the transportation of
summer water vapor from the Indian and Pacific Oceans to eastern CA. Additionally,
Huang et al. (2015b) stated that increased summer precipitation in the Tarim Basin is
mainly related to the weakened Indian summer monsoon. In addition, the large-scale
topography, such as the Qinghai-Tibet Plateau, causes the westerlies to flow around the
plateau rather than over it, which in turn influences the local transport of water vapor
and results in local precipitation changes (Xie et al., 2014). Therefore, the atmospheric
circulation and topographic factors bear on the transportation and content of water
vapor at short-term timescales, which makes the east of CA with summer precipitation
regime different from the core region of CA, but linked to the EA.

### 4.2 Possible dynamics of seasonal signals at long-term timescales

Studying the mechanism of paleoclimate change in EA and CA during the LGM
and MH, with model simulation, is of great significance for assessing future
hydroclimate changes. The results of paleoclimate simulations explain the difference
and linkage in the dry/wet status from EA and CA under the framework of seasonal
signals at long-term timescales. During the LGM, lower summer insolation increases
the meridional temperature difference and sea level pressure in the summer largely (Fig.
8k; 9a and c), leading to the strengthening of the westerlies (Fig. 10a) and further



increasing precipitation in the core region of CA (Fig. 9e). Given the weakening of the
LGM summer monsoon and the complex control factors (Fig. 10c), however, the
summer precipitation in the east of CA is weaker than that of MH (Fig. 9e), which is
consistent with the dry/wet status in EA and reflects the linkage between EA and the
east of CA caused by the summer precipitation regime. Although the westerlies weaken
in the LGM winter (Fig. 10b), the higher winter insolation contributes to the general
warming in CA and EA (Fig. 8k; 9b), resulting in lower relative humidity (Fig. 9d).
According to climatological theory (Barry and Richard, 2009), the decrease in relative
humidity means the increase in saturated water vapor pressure, which ultimately leads
to the increasing precipitation (Fig. 9f). Therefore, this elaborates the asynchrony of the
long-term dry/wet status in EA and CA under the control of seasonal signals.

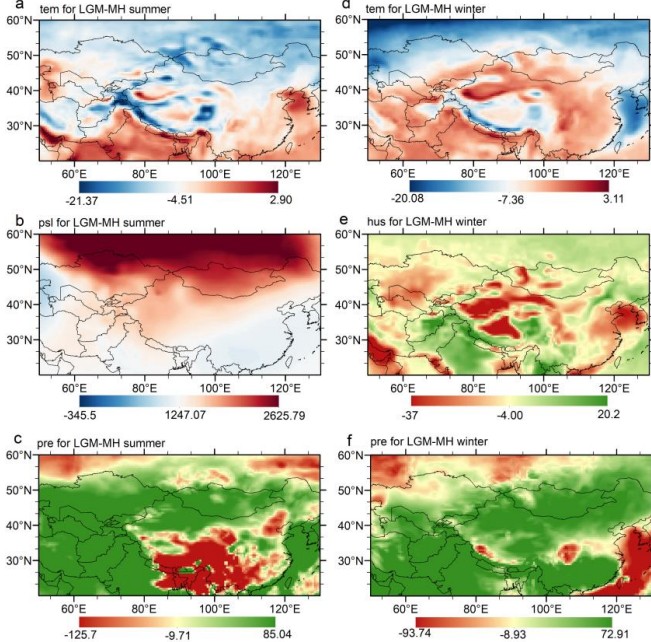


**Figure. 9** Summer differences of temperature (tem) (a), sea level pressure (psl) (c), precipitation (pre) (e), 700 hPa
wind field (g), and 200 hPa wind field (h) for LGM-MH; and winter differences of temperature (b), relatively humid



(hus) (d), precipitation (f), and 200 hPa wind field (i) for LGM-MH in EA and CA based on the PMIP3-CMIP5
multi-model ensemble.
Investigating the past climate is key to informing future climate change (Tierney
et al., 2020). From the perspective of paleoclimatology, monsoon and westerlies vary
greatly between LGM and MH, modulated by primary forces such as orbital insolation,
greenhouse gas, and ice sheets (Oster et al., 2015; Bereiter et al., 2015; Sime et al.,
2016). Paleoclimate records clearly indicate the wet status during the LGM and LH in
CA and the MH wet in EA (Fig. 8). Specifically, the dry/wet status in CA, affected by
the westerlies and characterized by wet climate conditions during the LGM and mid-
and late-Holocene, is opposite to that in monsoon-dominated EA. However, the proxy
records in CA similar to the monsoon evolution are located in the modern summer
precipitation region. From the perspective of precipitation seasonality, there are two
different precipitation regimes within CA. The core region of CA has a Mediterranean
climate (winter precipitation regime), with a dry summer and with seasonal
precipitation from early winter to late spring (Fig. 1); whereas, in the east of CA,
including northwest of China and west and south of Mongolia, the summer precipitation
contributes more (summer precipitation regime; Fig. 1). Therefore, summer
precipitation regime may be a potential forcing factor for the linkage of paleoclimate
reconstruction between EA and the east of CA, and the difference in precipitation
regime may result in a divergent moisture history in EA and the core region of CA.

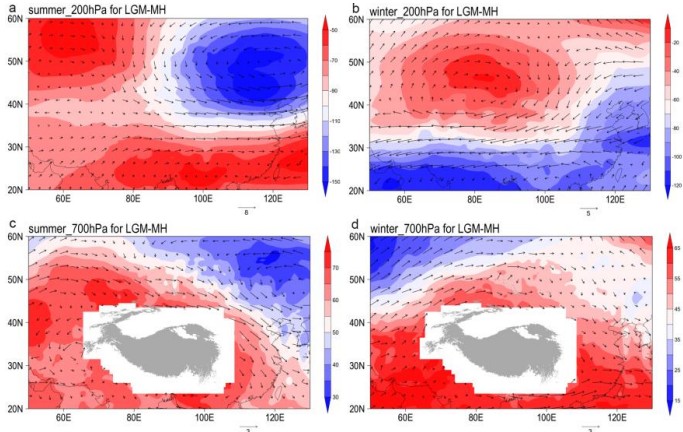

**Figure. 10** Summer differences of 200 hPa wind field (a) and 700 hPa wind field (c) for LGM-MH; and winter differences of 200 hPa wind field (b) and 700 hPa wind field (d) for LGM-MH in EA and CA based on the PMIP3-CMIP5 multi-model ensemble.

As a whole, our results provide a hypothesis that seasonal signals of precipitation derived from the simultaneity of rain and heat periods govern the difference and linkage in dry/wet status from EA and CA at multi-time scales. With recent global warming, some recent work also points out increasing summer precipitation in arid CA (Chen et al., 2021a; Ren et al., 2022). Meanwhile, the phenomenon of warmer and wetter climates coincides with the simultaneity of rain and heat periods (Hu and Han, 2022). Future work should focus on the fusion of multiple datasets and high-precision climate simulation designed to evaluate the mechanism.

## 5. Conclusion

The summer precipitation regime in EA and the east of CA and the winter precipitation regime in the core region of CA reveal seasonal signals of precipitation. Using the EOF method, this study analyzes the spatiotemporal variations of precipitation in EA and CA. Results reveal that seasonal signals derived from the



simultaneity of rain and heat periods are important factors linking climate change
modes in EA and CA at short-term timescales. A compilation of 42 proxy records with
reliable chronologies enables us to reassess the dry/wet status in EA and CA since the
LGM. Concurrently, paleoclimate records reflect seasonal signals triggered by the
insolation at long-term timescales. The multi-model simulations of multiple climatic
elements explain the climate mechanism of differences and linkage in dry/wet status
from EA and CA. In the traditional context of asynchronous dry/wet status between
summer precipitation regions and winter precipitation regions, we believe that regional
linkages also exist in EA and CA affected by seasonal signals.

## Acknowledgment
This research was supported by the National Natural Science Foundation of China
(Grant No. 42077415); the Second Tibetan Plateau Scientific Expedition and Research
Program (STEP) (Grant No. 2019QZKK0202); the Strategic Priority Research Program
of Chinese Academy of Sciences (Grant No. XDA20100102); the 111 Project
(BP0618001).

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

Monitoring global drought using the self-calibrating Palmer Drought Severity
Index [in \State of the Climate in 2020\]. Bulletin of the American Meteorological





487  Society 101. S51-S52.

488 Barry, R.G., Richard, J.C., 2009. Atmosphere, weather and climate. Routledge.

489 Bereiter, B., Eggleston, S., Schmitt, J., Nehrbass‑Ahles, C., Stocker, T. F., Fischer, H.,
490  Kipfstuhl, S., Chappellaz, J., 2015. Revision of the EPICA Dome C CO2 record
491  from 800 to 600 kyr before present. Geophys. Res. Lett. 42, 542-549.

492 Blyakharchuk, T.A., Wright, H.E., Borodavko, P.S., van der Knaap, W.O., Ammann, B.,
493  2007. Late glacial and Holocene vegetational history of the Altai mountains
494  (southwestern tuva republic, siberia). Palaeogeogr. Palaeoclimatol. Palaeoecol.
495  245, 518-534.

496 Briegleb, B.P., Bitz, C.M., Hunke, E.C., Lioscomb, W.H., Holland, M.M., Schramm,
497  J.L., Moritz, A.R., 2004. Scientific description of the sea ice component in the
498  community climate system model. Version, 3, 70.

499 Cai, Y., Tan, L., Cheng, H., An, Z., Edwards, R. L., Kelly, M. M., Kong, X., Wang, X.,
500  2010. The variation of summer monsoon precipitation in central China since the
501  last deglaciation. Earth Planet Sci. Lett. 291, 21-31.

502 Chen, C., Zhang, X., Lu, H., Jin, L., Du, Y., Chen, F., 2021a. Increasing summer
503  precipitation in arid central Asia linked to the weakening of the East Asian summer
504  monsoon in the recent decades. Int. J. Climatol. 41, 1024-1038.

505 Chen, F., Chen, J., Huang, W., 2021b. Weakened East Asian summer monsoon triggers
506  increased precipitation in Northwest China. Sci. China Earth Sci. 64, 835-837.

507 Chen, F., Chen, J., Huang, W., Chen, S., Huang, X., Jin, L., Jia, J., Zhang, X., An, C.,
508  Zhang, J., Zhao, Y., 2019. Westerlies Asia and monsoonal Asia: spatiotemporal
509  differences in climate change and possible mechanisms on decadal to suborbital
510  timescales. Earth Sci. Rev. 192, 337-354.

511 Chen, F., Jia, J., Chen, J., Li, G., Zhang, X., Xie, H., Xia, D., Huang, W., An, C., 2016.
512  A persistent Holocene wetting trend in arid central Asia, with wettest conditions
513  in the late Holocene, revealed by multi-proxy analyses of loess-paleosol sequences
514  in Xinjiang, China. Quat. Sci. Rev. 146, 134-146.

515 Chen, F., Chen, J., Huang, W., 2009. A discussion on the westerly-dominated climate
516  model in mid-latitude Asia during the modern interglacial period. Earth Sci. Front.
517  16, 23-32 (in Chinese with English abstract).

518 Chen, F., Yu, Z., Yang, M., Ito, E., Wang, S., David, B.M., Huang, X., Zhao, Y., Sato,
519  T., Birks, H.J.B., Boomer, I., Chen, J., An, C., Wünnemann, B., 2008. Holocene
520  moisture evolution in arid Central Asia and its out-of-phase relationship with
521  Asian monsoon history. Quat. Sci. Rev. 27, 351-364.

522 Chen, G., Huang, R., 2012. Excitation mechanisms of the tele-connection patterns
523  affecting the July precipitation in North-west China. J. Clim. 25, 7834-7851.

524 Chen, S., Chen, J., Lv, F., Liu, X., Huang, W., Wang, T., Liu, J., Hou, J., Chen, F., 2022.
525  Holocene moisture variations in arid central Asia: Reassessment and reconciliation.
526  Quat. Sci. Rev. 297, 107821.

527 Chen, S., Liu, J., Wang, X., Zhao, S., Chen, J., Qiang, M., Liu, B., Xu, Q., Xia, D.,
528  Chen, F., 2021c. Holocene dust storm variations over northern China: transition
529  from a natural forcing to an anthropogenic forcing. Sci. Bull. 66, 2516-2527.

530 Cheng, H., Sp€otl, C., Breitenbach, S.F.M., Sinha, A., Wassenburg, J.A., Jochum, K.P.,





Scholz, D., Li, X.L., Peng, Y.B., Lv, Y.B., Zhang, P.Z., Votintseva, A., Loginov, V.,
Ning, Y.F., Kathayat, G., Edwards, R.L., 2016. Climate variations of Central Asia
on orbital to millennial timescales. Sci. Rep. 6, 36975.
Cheng, H., Zhang, P., Sp€otl, C., Edwards, R.L., Cai, Y., Zhang, D., Sang, W., Tan, M.,
An, Z., 2012. The climatic cyclicity in semiarid-arid central Asia over the past
500,000 years. Geophys. Res. Lett. 39, L01705.
Dykoski, C.A., Edwards, R.L., Cheng, H., Yuan, D., Cai, Y., Zhang, M., Lin, Y., Qing,
538        J., An, Z., Revenaugh, J., 2005. A high-resolution, absolute-dated Holocene and
deglacial Asian monsoon record from Dongge Cave, China. Earth Planet. Sci. Lett.
233, 71-86.
Feng, Z.D., Sun, A.Z., Abdusalih, N., Ran, M., Kurban, A., Lan, B., Zhang, D.L., Yang,
Y.P., 2017. Vegetation changes and associated climatic changes in the southern
Altai Mountains within China during the Holocene. Holocene 27, 683-693.
Gao, F.Y., Jia, J., Xia, D.S., Lu, C.C., Lu, H., Wang, Y.J., Liu, H., Ma, Y.P., Li, K.M.,
2019. Asynchronous Holocene climate optimum across mid-latitude Asia.
Palaeogeogr. Palaeoclimatol. Palaeoecol. 518, 206e214.
Gent, P.R., Danabasoglu, G., Donner, L.J., Holland, M.M., Hunke, E.C., Jayne, S.R.,
Lawrence, D.M., Neale, R.B., Rasch, P.J., Vertenstein, M., 2011. The community
climate system model version 4. J. Clim. 24, 4973-4991.
Guan, X., Yang, L., Zhang, Y., & Li, J. (2019). Spatial distribution, temporal variation,
and transport characteristics of atmospheric water vapor over Central Asia and the
arid region of China. Global Planet. Change 172, 159-178.
Han, S.T., Wu, N.Q., Li, Z.Z., 1993. Inland climate changes in Dzungaria during the
late Pleistocene Epoch. Geographical Research. 12, 47-54 (in Chinese with
English abstract).
Han, S.T., Qu, Z., 1992. Inland Holocene climatic features recorded in Balikun lake,
northern Xinjiang. Science in China Series B-Chemistry, Life Sciences & Earth
Sciences (in Chinese) 11, 1201-1209.
Harris, I., Jones, P.D., Osborn, T.J., Lister, D.H., 2014. Updated high-resolution grids
of monthly climatic observations-the CRU TS3.10 Dataset. Int. J. Climatol. 34,
623-642.
Heinecke, L., Mischke, S., Adler, K., Barth, A., Biskaborn, B.K., Plessen, B., Ingmar,
563        N., Gerhard, K., Ilhomjon, R., Herzschuh, U., 2017. Climatic and limnological
changes at Lake Karakul (Tajikistan) during the last ~29 cal ka. J. Paleolimnol. 58,
317-334.
Hersbach, H., Bell, B., Berrisford, P., Hirahara, S., Horányi, A., Muñoz-Sabater, J.,
Nicolas, J., Peubey, C., Radu, R., Schepers, D., Simmons, A., Soci, C., Abdalla, S.,
Abellan, X., Balsamo, G., Bechtold, P., Biavati, G., Bidlot, J., Bonavita, M., . . .
Hersbach, H., 2020. The ERA5 global reanalysis. Quarterly Journal of the Royal
Meteorological Society, 146, 1999-2049.
Herzschuh, U., 2006. Palaeo-moisture evolution in monsoonal central Asia during the
last 50,000 years. Quat. Sci. Rev. 25, 163-178.
Hu, Q., Han, Z.H., 2022. Northward Expansion of Desert Climate in Central Asia in
Recent Decades. Geophys. Res. Lett. 49, e2022GL098895.





Huang, W., Chen, J.H., Zhang, X.J., Feng, S., Chen, F.H., 2015a. Definition of the core
zone of the "westerlies-dominated climatic regime", and its controlling factors
during the instrumental period. Sci. China Earth Sci. 58, 676-684.

Huang, W., Feng, S., Chen, J. and Chen, F. 2015b. Physical mechanisms of summer
precipitation variations in the Tarim basin in northwestern China. J. Clim. 28,
3579-3591.

Huang, X.Z., Chen, F.H., Fan, Y.X., Yang, M.L., 2009. Dry late-glacial and early
Holocene climate in arid central Asia indicated by lithological and palynological
evidence from Bosten Lake, China. Quat. Int. 194, 19-27.

Hurrell, J.W., Deser, C., 2009. North Atlantic climate variability: The role of the North
Atlantic Oscillation. J. Mar. Syst., 78, 28-41.

Hurrell, J.W., 1995. Decadal Trends in the North Atlantic Oscillation: Regional
Temperatures and Precipitation. Science 269, 676-679.

Jia, J., Chen, J.H., Wang, Z.Y., Chen, S.Q., Wang, Q., Wang, L.B., Yang, L.W., Xia,
D.S., Chen, F.H., 2021. No evidence for an anti-phased Holocene moisture regime
in mountains and basins in Central Asian: records from Ili loess, Xinjiang.
Palaeogeogr. Palaeoclimatol. Palaeoecol. 572, 110407.

Jiang, Q.F., Ji, J.F., Shen, J., Matsumoto, R., Tong, G.B., Qian, P., Ren, X.M., Yan, D.Z.,
2013. Holocene vegetational and climatic variation in westerly-dominated areas
of Central Asia inferred from the Sayram Lake in northern Xinjiang, China. Sci.
China Earth Sci. 56, 339-353.

Jiang, Q.F., Meng, B.W., Wang, Z., Qian, P., Zheng, J.N., Jiang, J.W., Zhao, C., Hou,
597        J.Z., Dong, G.W., Shen, J., Liu, W.G., Liu, Z.H., Chen, F.H., 2022. Exceptional
terrestrial warmth around 4200-2800 years ago in Northwest China. Sci. Bull. 67,
427-436.

Joussaume, S., Taylor, K.E., Braconnot, P.J.F.B., Mitchell, J.F.B., Kutzbach, J.E.,
Harrison, S.P., Prentice, I.C., Broccoli, A.J., Abe-Ouchi, A., Bartlein, P.J., Bonfils,
C., 1999. Monsoon changes for 6000 years ago: results of 18 simulations from the
paleoclimate modeling intercomparison project (PMIP). Geophys. Res. Lett. 26,
859-862.

Kang, S.G., Wang, X.L., Roberts, H.M., Duller, G.A., Song, Y.G., Liu, W.G., Zhang,
R., Liu, X.X., Lan, J.H., 2020. Increasing effective moisture during the Holocene
in the semiarid regions of the Yili Basin, central Asia: evidence from loess sections.
Quat. Sci. Rev. 246, 106553.

Lee, M.K., Lee, Y.I., Lim, H.S., Lee, J.I., Yoon, H.I., 2013. Late Pleistocene-H olocene
records from Lake Ulaan, southern Mongolia: implications for east Asian
palaeomonsoonal climate changes. J. Quat. Sci. 28, 370-378.

Leroy, S.A., López-Merino, L., Tudryn, A., Chalié, F., Gasse, F., 2014. Late Pleistocene
and Holocene palaeoenvironments in and around the middle Caspian basin as
reconstructed from a deep-sea core. Quat. Sci. Rev. 101, 91-110.

Leroy, S.A.G., Ricketts, R.D., Rasmussen, K.A., 2021. Climatic and limnological
changes 12,750 to 3600 years ago in the Issyk-Kul catchment, Tien Shan, based
on palynology and stable isotopes. Quat. Sci. Rev. 259, 106897.

Leroy, S.A.G., Lopez-Merino, L., Tudryn, A., Chalie, F., Gasse, F., 2014. Late



Pleistocene and Holocene palaeoenvironments in and around the middle Caspian
basin as reconstructed from a deep-sea core. Quat. Sci. Rev. 101, 91-110.
Li, J.J., 1990. The patterns of environmental changes since last Pleistoc ene in
northwestern China. Quat. Sci. 3, 197-204 (in Chinese with English abstract).
Li, X.Q., Zhao, K.L., Dodson, J., Zhou, X.Y., 2011. Moisture dynamics in central Asia
for the last 15 kyr: new evidence from Yili Valley, Xinjiang, NW China. Quat. Sci.
Rev. 30, 3457-3466.
Li, Y., Song, Y.G., Kaskaoutis, D.G., Zan, J.B., Orozbaev, R., Tan, L.C., Chen, X.L.,
2021. Aeolian dust dynamics in the Fergana Valley, Central Asia, since~ 30 ka
inferred from loess deposits. Geosci. Front. 12, 101180.
Li, Y., Song, Y.G., Orozbaev, R., Dong, J.B., Li, X.Z., Zhou, J., 2020a. Moisture
evolution incentral Asia since 26 ka: insights from a Kyrgyz loess section, western
tian Shan. Quat. Sci. Rev. 249, 106604.
Li, Y., Peng, S., Liu, H., Zhang, X., Ye, W., Han, Q., Zhang, Y., Xu, L., Li, Y.C., 2020b.
Westerly jet stream controlled climate change mode since the Last Glacial
Maximum in the northern Qinghai-Tibet Plateau. Earth and Planetary Science
Letters, 549, 116529.
Lioubimtseva, E., 2014. Impact of Climate Change on the Aral Sea and its Basin. The
Devastation and Partial Rehabilitation of a Great Lake, The Aral Sea, pp. 405-427.
Liu, X.Q., Shen, J., Wang, S.M., Wang, Y.B., Liu, W.G., 2007. Southwest monsoon
changes indicated by oxygen isotope of ostracode shells from sediments in
Qinghai lake since the late glacial. Chin. Sci. Bull. 4, 109-114.
Liu, X.K., Liu, J.B., Shen, C.C., Yang, Y., Chen, J.H., Chen, S.Q., Wang, X.F., Wu, C.C.,
Chen, F.H., 2020. Inconsistency between records of $\delta^{18}O$ and trace element ratios
from stalagmites: evidence for increasing midelate Holocene moisture in arid
central Asia. Holocene 30, 369-379.
Lorenz, E.N., 1956. Empirical orthogonal function and statistical weather prediction.
Scientific Report No. 1 Statist Forecasting Project. Department of Meteorology,
Massachusetts Institute of Technology.
Manoj, M.C., Srivastava, J., Uddandam, P.R., Thakur, B., 2020. A 2000 year multiproxy
evidence of natural/anthropogenic influence on climate from the southwest coast
of India. J. Earth Sci. 31, 1029-1044.
Mantua, N.J., Hare, S.R., 2002. The Pacific Decadal Oscillation. J. Oceanogr. 58, 35-
44.

Mischke, S., Lai, Z.P., Aichner, B., Heinecke, L., Makhmudov, Z., Kuessner, M.L.,
Herzschuh, U., 2017. Radiocarbon and optically stimulated luminescence dating
of sediments from Lake Karakul, Tajikistan. Quat. Geochronol. 41, 51-61.
Nagashima K, Tada R, Tani A, et al. 2011. Millennial-scale oscillations of the westerly
jet path during the last glacial period. Journal of Asian Earth Sciences. 40 1214-
1220. https://doi.org/10.1016/j.jseaes.2010.08.010.
Peltier, W.R., 2004. Global glacial isostasy and the surface of the ice-age Earth: The
ICE-5G (VM2) model and GRACE. Annu. Rev. Earth Planet. Sci. 32, 111-149.
Peng, D., Zhou, T., 2017. Why was the arid and semiarid North-west China getting
wetter in the recent decades? J. Geophys. Res. Atmos. 122, 9060-9075.



Ran, M., Feng, Z.D., 2014. Variation in carbon isotopic composition over the past ca.
46,000 yr in the loessepaleosol sequence in central Kazakhstan and paleoclimatic
significance. Org. Geochem. 73, 47-55.
Randall, D.A., Wood, R.A., Bony, S., Colman, R., Taylor, K.E., 2007. Climate models
and their evaluation. In Climate change 2007: The physical science basis.
Contribution of Working Group I to the Fourth Assessment Report of the IPCC
(FAR) (pp. 589-662). Cambridge University Press.
Rayner, N. A. A., Parker, D. E., Horton, E. B., Folland, C. K., Alexander, L. V., Rowell,
D. P., Kent, E.C., 2003. Global analyses of sea surface temperature, sea ice, and
night marine air temperature since the late nineteenth century, J. Geophys. Res.
Atmos. 108, 4407.
Ren, Y., Yu, H., Liu, C., He, Y., Huang, J., Zhang, L., Hu, H., Zhang, Q., Chen, S., Liu,
X., Zhang, M., Wei, Y., Yang, Y., Fan, W., Zhou, J., 2021. Attribution of Dry and
Wet Climatic Changes over Central Asia. J. Clim. 35,1399-1421.
Ricketts, R.D., Johnson, T.C., Brown, E.T., Rasmussen, K.A., Romanovsky, V.V., 2001.
The Holocene paleolimnology of Lake Issyk-Kul, Kyrgyzstan: trace element and
stable isotope composition of ostracodes. Palaeogeogr. Palaeoclimatol. Palaeoecol.
176, 207-227.
Rotstayn, L., Collier, M., Dix, M., Feng, Y., Gordon, H., O'Farrell, S., Smith, I., Syktus,
682        J., 2010. Improved simulation of Australian climate and ENSO-related climate
variability in a GCM with an interactive aerosol treatment. Int. J. Climatol. 30,
1067-1088.
Rudaya, N., Tarasov, P., Dorofeyuk, N., Solovieva, N., Kalugin, I., Andreev, A., Daryin,
686        A., Diekmann, B., Riedel, F., Tserendash, N., Wagner, M., 2009. Holocene
environments and climate in the Mongolian Altai reconstructed from the Hoton-
Nur pollen and diatom records: a step towards better understanding climate
dynamics in Central Asia. Quat. Sci. Rev. 28, 540-554.
Schmidt GA, Kelley M, Nazarenko L, Ruedy R, Russell GL, Aleinov I, Bauer M, Bauer
SE, Bhat MK, Bleck R, Canuto V, Chen YH, Cheng Y, Clune TL, Genio AD, de
Fainchtein R, Faluvegi G, Hansen JE, Healy RJ, Kiang NY, Koch D, Lacis AA,
LeGrande AN, Lerner J, Lo KK, Matthews EE, Menon S, Miller RL, Oinas V,
Oloso AO, Perlwitz JP, Puma MJ, Putman WM, Rind D, Romanou A, Sato M,
Shindell DT, Sun S, Syed RA, Tausnev N, Tsigaridis K, Unger N, Voulgarakis A,
Yao MS, Zhang JL. 2014. Configuration and assessment of the GISS ModelE2
contributions to the CMIP5 archive. Journal of Advances in Modeling Earth
Systems. 6, 141-184.
Sun, A., Feng, Z., Ran, M., Zhang, C., 2013. Pollen-recorded bioclimatic variations of
the last ~22,600 years retrieved from Achit Nuur core in the western Mongolian
Plateau. Quat. Int. 311, 36-43.
Tao, S.C., An, C.B., Chen, F.H., Tang, L.Y., Wang, Z.L., Lü, Y.B., Li, Z.F., Zheng, T.M.,
Zhao, J.J., 2010. Pollen-inferred vegetation and environmental changes since 16.7
704        ka BP at Balikun Lake, Xinjiang. Chin. Sci. Bull. 55, 2449-2457.
Tian, F., Herzschuh, U., Telford, R. J., Mischke, S., Van der Meeren, T., Krengel, M.,
2014. A modern pollen-climate calibration set from central‐western Mongolia



and its application to a late glacial-Holocene record. J. Biogeogr. 41, 1909-1922.

Tierney, J.E., Poulsen, C.J., Montañez, I P et al., 2020. Past climates inform our future. Science, 370(6517), 3701–3709.

Tierney, J.E., Poulsen, C.J., Montañez, I.P., Bhattacharya, T., Feng, R., Ford, H.L., Hönisch, B., Inglis, G.N., Petersen, S.V., Sagoo, N., Tabor, C. R., Thirumalai, K., Zhu, J., Burls, N.J., Foster, G.L., Goddéris, Y., Huber, B.T., Ivany, L.C., Turner, S., Lunt, D.J., Mcelwain, J.C., Mills. B.J.W., Otto-Bliesner, B.L., Ridgwell, A., Zhang, Y., 2020. Past climates inform our future. Science, 370, eaay3701.

Voldoire, A., Sanchez-Gomez, E., Mélia, D.S.Y., Decharme, B., Cassou, C., Sénési, S., et al. 2013. The CNRM-CM5.1 global climate model: Description and basic evaluation. Clim. Dyn. 40, 2091-2121.

Wang, B., Liu, J., Kim, H.J., Webster, P.J., Yim, S.Y., 2012. Recent change of the global monsoon precipitation (1979-2008). Clim. Dyn. 39, 1123-1135.

Wang, L., Jia, J., Xia, D., Liu, H., Gao, F., Duan, Y., Wang, Q., Xie, H., Chen, F., 2018. Climate change in arid central Asia since MIS 2 revealed from a loess sequence in Yili Basin, Xinjiang, China. Quat. Int. 502, 258-266.

Wang, P., Wang, B., Cheng, H., Fasullo, J., Guo, Z., Kiefer, T., Liu, Z., 2017. The global monsoon across time scales: Mechanisms and outstanding issues. Earth Sci. Rev. 174, 84-121.

Wang, Y.J., Cheng, H., Edwards, R.L., An, Z.S., Wu, J.Y., Shen, C.C., Dorale, J.A., 2001. A high-resolution absolute-dated late Pleistocene monsoon record from Hulu cave, China. Science 294, 2345-2348.

Wang, Q., Wei, H.T., Khormali, F., Wang, L.B., Xie, H.C., Wang, X., Huang, W., Chen, J.H., Chen, F.H., 2020. Holocene moisture variations in western arid central Asia inferred from loess records from NE Iran. G-cubed 21, e2019GC008616.

Watanabe, S., Hajima, T., Sudo, K., Nagashima, T., Takemura, T., Okajima, H., Nozawa, T., Kawase, H., Abe, M., Yokohata, T., Ise, T., Sato, H., Kato, E., Takata, K., Emori, S., Kawamiya, M., 2011. MIROC-ESM 2010: model description and basic results of CMIP5-20c3m experiments. Geoscientific Model Development. 4, 845-872,

Wei, W., Zhang, R., Wen, M., Yang, S., 2017. Relationshipbetween the Asian westerly jet stream and summer rainfall over Central Asia and North China: roles of the Indian mon-soon and the south Asian high. J. Clim. 30, 537-552.

Wu, P., Ding, Y., Liu, Y., Li, X., 2019. The characteristics of moisture recycling and its impact on regional precipitation against the background of climate warming over Northwest China. Int. J. Climatol. 39, 5241-5255.

Xie, C.Y., Li, M.J., Zhang, X.Q., 2014. Characteristics of summer atmospheric water resources and its causes over the Tibetan plateau in recent 30 years. J. Nat. Resour. 29, 979-989.

Xu, H., Lan, J., Zhang, G., Zhou, X., 2019. Arid Central Asia saw mid-Holocene drought. Geology 47, 255-258.

Yang, Y.P., Feng, Z.D., Zhang, D.L., Lan, B., Ran, M., Wang, W., Sun, A.Z., 2021. Holocene hydroclimate variations in the eastern Tianshan Mountains of northwestern China inferred from a palynological study. Palaeogeogr. Palaeoclimatol. Palaeoecol. 564, 110184.



Yu, G., Xue, B., Wang, S.M., et al., 2000. Chinese lakes records and the climate significance during Last Glacial Maximum. Chin. Sci. Bull. 45, 250-255 (in Chinese with English abstract).

Yukimoto, S., Adachi, Y., Hosaka, M., Sakami, T., Yoshimura, H., Hirabara, M., Tanaka, T.Y., Shindo, E., Tsujino, H., Deushi, M., 2012. A new global climate model of the Meteorological Research Institute: MRI-CGCM3: Model description and basic performance. J. Meteorol. Soc. Japan. 90, 23-64.

Zhang, H.L., Yu, K.F., Zhao, J.X., Feng, Y.X., Lin, Y.S., Zhou, W., Liu, G.H., 2013. East Asian Summer Monsoon variations in the past 12.5 ka: High-resolution δ18O record from a precisely dated aragonite stalagmite in central China. J. Asian Earth Sci. 73, 162-175.

Zhang, J., Nottebaum, V., Tsukamoto, S., Lehmkuhl, F., Frechen, M., 2015. Late Pleistocene and Holocene loess sedimentation in central and western Qilian Shan (China) revealed by OSL dating. Quat. Int. 372, 120-129.

Zhang, J.C., Lin, Z.G., 1992. Climate of China. Wiley, New York.

Zhang, D.L., Chen, X., Li, Y.M., Ran, M., Yang, Y.P., Zhang, S.R., Feng, Z.D., 2020. Holocene moisture variations in the arid central Asia: new evidence from the southern Altai mountains of China. Sci. Total Environ. 735, 139545.

Zhang, D.L., Feng, Z.D., 2018. Holocene climate variations in the Altai Mountains and the surrounding areas: a synthesis of pollen records. Earth Sci. Rev. 185, 847-869.

Zhang, Q., Lin, J., Liu, W., Han, L., 2019. Precipitation seesaw phenomenon and its formation mechanism in the eastern and western parts of Northwest China during the flood season. Sci. China Earth Sci. 62, 2083-2098.

Zhao, Y., An, C., Mao, L., Zhao, J., Tang, L., Zhou, A., Li, H., Dong, W., Duan, F., Chen, F., 2015. Vegetation and climate history in arid western China during MIS2: New insights from pollen and grain-size data of the Balikun Lake, eastern Tien Shan. Quat. Sci. Rev. 126, 112-125.

Sorg, A., Bolch, T., Stoffel, M., Solomina, O., Beniston, M., 2012. Climate change impacts on glaciers and runoff in Tien Shan (Central Asia). Nat Clim Change. 2, 725-731.

Zhang, D.L., Feng, Z.D., 2018. Holocene climate variations in the Altai Mountains and the surrounding areas: A synthesis of pollen records. Earth Sci Rev. 185, 847-869.