# Peer review of "Simultaneous seasonal dry/wet signals in Eastern and Central Asia since the Last Glacial Maximum"

_Climate of the Past, 2023_

## Author Comment (AC1)

Dear Editor and reviewers,

Thank you for your letter and for the reviewer's comments concerning our manuscript entitled "Tracing seasonal signals in dry/wet status for regions with simultaneous rain and heat from Eastern and Central Asia since the Last Glacial Maximum". Thanks very much for giving us such a precious chance. Those comments and suggestions are all valuable and very helpful for revising and improving our paper, as well as the important guiding significance to our research. We have studied all of the comments and suggestions carefully and made corresponding corrections in the manuscript with colored text. A detailed list of revisions against each point of reviewer 1 that is being raised is below.

**Reviewer 1**

This manuscript presents a comprehensive study of the hydroclimate evolution of the central and eastern Asia region since the Last Glacial Maximum (LGM). The authors analyze the similarities and differences between the spatiotemporal hydroclimate regimes in these regions, at both modern and paleoclimate timescales, using modern climate datasets, modelling data, and a compilation of more than forty paleoclimate records. The manuscript elaborates on the asynchronous nature of short-term and long-term dry/wet patterns in central and eastern Asia. Modern climate regimes in these two regions show a strong anti-phase climate pattern, marked by the westerly-dominated "winter" regime in central Asia and the monsoon-dominated "summer" regime in eastern Asia. However, previous paleoclimate reconstructions suggest that these spatiotemporal differences could have changed during late Pleistocene and the Holocene periods, especially towards the eastern border of central Asia. With their new statistical analyses on modern climate data, summed to an exhaustive revision of published proxy data, the author adds interesting information that complements well with other published studies, proposing that monsoonal precipitation might be key to understanding the evolution of these two climate regimes since the LGM.

However, the article could, in my opinion, be improved in several ways. Here I summarize my suggestions for major changes and several other minor amendments to the text and figures.

Response: We'd like to thank your constructive comments and suggestions on our manuscript. We appreciate the chance to clarify some concepts and have adopted all suggested changes raised by Reviewer 1. In addition, we have scrutinized the text for additional minor errors and to improve clarity and readability, which are also highlighted in the revised version. We sincerely hope this revision meets with your approval.

**Conceptualization**

I think that the authors do not provide enough information to clearly state why exploring the past and present spatiotemporal changes in the two regions is scientifically relevant. Although differences in the interpretation of paleoclimate data exist, why does this merit to analyze modern climate timeseries and compile such a great number of paleoclimate records? Also, I personally miss an explanation to why combining modern data analysis with paleoclimate records and past climate simulations is a valid avenue to "assessing future hydroclimate changes" (line 403). It would be great if these issues can be addressed in a new version of the introduction.

Response: Thanks very much for your suggestions and carefulness. Firstly, because the concept of simultaneity of rain and heat period is derived from the modern climate, we first start from the modern climate to analyze the impact of simultaneous rain and heat on the differences and linkages in precipitation between CA and EA. We believe this can improve the reliability and scientific for applying this concept to paleoclimate analysis. At the same time, jointly analyzing modern climate characteristics and paleoclimate history can more comprehensively explore the impact of seasonal signals of precipitation on dry/wet status in CA and EA. Secondly, we have deleted the inappropriate description (e.g. assessing future hydroclimate changes in line 403) and the updated sentence is "Model simulations are a valid means to visually study the mechanism of paleoclimate change in EA and CA during the LGM and MH.".

Also, I found it troubling to understand some concepts used throughout the manuscript. For instance, what does "wet/dry status" mean (e.g., line 74)? In my opinion, the authors should be more careful in placing complex terminology without further explanations.

Response: Thanks very much for your suggestions. We have checked all the complex terminology in the text and added more explanations. The description in line 74 is amended to "In monsoonal EA, a strengthened summer monsoon and humid climate usually occurs in the early and mid Holocene, and a weakened summer monsoon and drier climate prevailed during the late Holocene (Dykoski et al., 2005; Chen et al., 2015). Based on the integration of paleoclimate records, modern meteorological observation data and paleoclimate simulations, Chen et al. (2008, 2009, 2019) revealed that the 'westerlies-dominated climatic regime' in arid CA presents dry early Holocene, wetter mid-Holocene, and moderately wet late Holocene, which is out-of-phase or anti-phased with the dry/wet status in the monsoon-dominated regions.".

**Abstract**

With 393 words, the abstract is too long in my opinion and includes not only the most essential results and discussions presented in the manuscript. Phrases such as "The remarkable difference in summer precipitation regime and winter precipitation regime reveal the seasonal signals of precipitation in multi-time scale climate change." (Lines 17-19) and "In particular, summer and winter precipitation in EA and CA is associated with the Asian monsoon, westerlies, ENSO, NAO, and PDO" (lines 30-32) do not provide any critical information and are too general for an abstract. I would favor a shorter and concise abstract.

Response: Thanks very much for your suggestion. We have shortened the abstract and the updated contents are reproduced below:
"

**Abstract**

The global monsoon region with the summer precipitation regime and the

Mediterranean climate region with the winter precipitation regime show opposite dry/wet evolution since the Last Glacial Maximum (LGM). Therefore, different precipitation regimes bring about the contradiction in dry/wet status between Eastern and Central Asia (EA and CA). Based on the comprehensive study of modern observation datasets, ensemble simulations of eight climate models from the Paleoclimate Model Intercomparison Project phase 3 (PMIP3), and the compilation of 42 proxy records from EA and CA, here we reveal that seasonal signals of precipitation derived from the simultaneity of rain and heat periods could govern the difference and linkage in dry/wet status from EA and CA. At short-term timescales, EOF analysis results of mean annual precipitation uncover different precipitation regimes in EA and CA. However, EOF results of summer and winter precipitation indicate the similarity between EA and the east of CA, suggesting that seasonal signals of precipitation affected by the Asian monsoon, westerlies, ENSO, NAO, and PDO are the primary factor causing the linkage in dry/wet status. At long-term timescales, proxy records since the LGM in EA and CA reveal parallel dry/wet evolution in EA and the east of CA as well. A visual inspection from PMIP3 multi-model simulations in summer and winter shows that the insolation in different seasons control the intensity of westerlies and summer monsoon and further influence the summer and winter precipitation in EA and CA since the LGM. Overall, we suggest, in addition to the traditional difference caused by different precipitation regimes, that dry/wet status in EA and CA universally have inter-regional connections affected by seasonal signals of precipitation at multi-time scales."

**Results**

A visual inspection of Figure 2a (the first EOF mode of mean annual precipitation during historical times) shows a spatial pattern which is quite different from the "simultaneous region of rain and heat periods" revealed in Figure 1. Hence, it is hard to justify that this EOF mode shows a clear "dipole mode" or a "seesaw pattern" as stated in section 3.1. Also, the variance contribution (10.3%) is clearly low, and therefore it is hard to argue that the pattern revealed by this EOF mode is a good representation of variance of the whole dataset. By contrast, a much clearer pattern emerges when this mode is decomposed in their seasonal components (Figure 3), and

when annual and winter-summer differences are analyzed (Figure 4). All this makes me think that Figure 2 is not contributing much to the data analysis. Perhaps a good alternative would be to add this Figure 1 to the supplements.

Response: Thanks very much for your suggestion. This is a good point. We have deleted some controversial descriptions and transferred Figure 2 to Supplemental materials.

**Conclusions**

The authors mention in the conclusions that "The multi-model simulations of multiple climatic elements explain the climate mechanism of differences and linkage in dry/wet status from EA and CA" (Lines 464-466), however there is no explanation as to what mechanisms they are talking about. Similarly, the conclusión mentions that "A compilation of 42 proxy records with reliable chronologies enables us to reassess the dry/wet status in EA and CA since the LGM.", but then again there is no mention about what this evolution of the dry/wet status has been. The conclusion does not provide any clear link between the results of the modern "short-term" EOF analysis and the descriptions of paleoclimate trends that comprises the "long-term" analysis. It would be great that the authors are able to address these issues in a new version of the manuscript.

Response: Thanks very much for your suggestions. We have added a detailed description for the climate mechanism and the evolution of the dry/wet status. Meanwhile, we have also clearly described the link between the modern "short-term" analysis and the paleoclimate "long-term" analysis. The updated conclusions are reproduced below:
"

**Conclusions**

The summer precipitation regime in EA and the east of CA and the winter precipitation regime in the core region of CA reveal seasonal signals of precipitation. Using the EOF method, this study analyzes the spatiotemporal variations of precipitation in EA and CA. Results reveal that seasonal signals derived from the

simultaneity of rain and heat periods are important factors linking climate change modes in EA and CA at short-term timescales. A compilation of 42 proxy records with reliable chronologies enables us to reassess the dry/wet status in EA and CA since the LGM. In core regions of CA, dry/wet status is usually characterized by dry EH and wet LH. However, most records in the east of CA with simultaneous rain and heat periods hold the same dry/wet status with EA, i.e., the dry condition during the LGM and the wet climate during the EH and MH. This also reflects another meaning of seasonal signals at long-term timescales, namely the "dry-cold" pattern and "wet-warm" pattern. Concurrently, paleoclimate records reflect seasonal signals triggered by the insolation at long-term timescales. The multi-model ensemble simulations of multiple climatic elements explain the climate mechanism of differences and linkage in dry/wet status from EA and CA since the LGM. Results show that summer insolation influences the meridional temperature gradient and sea level pressure in the summer, changing the intensity of the westerly winds and summer monsoon and further controlling the summer precipitation in EA and the east of CA. Meanwhile, winter insolation contributes to the general warming in EA and the core region of CA, and in turn results in lower relative humidity, which ultimately increases winter precipitation during the LGM.

In general, the seasonal signals of precipitation derived from the simultaneity and non-simultaneity of rain and heat periods on short-term timescales can also affect the dry/wet status on long-term timescales, but their influencing factors are different. Due to the influence of precipitation seasonal signals on multi-time scales, CA with the winter precipitation regime and EA with the summer precipitation regime show traditional anti-phase in the evolution of dry/wet status. However, it is worth noting that in the east of CA with simultaneous rain and heat, there is the same dry/wet evolution as in EA. Therefore, we believe that seasonal signals can provide important insight for analyzing the differences and linkages in climate change between CA and EA on multi-time scales.

"

**Minor changes**

Line 29: "…winter precipitation, suggested seasonal signals…".

I suggest adding "that" between after "suggested".

Response: Thanks very much for your carefulness. Done.

Line 30: I suggest removing the sentence "In particular, summer and winter precipitation in EA and CA is associated with the Asian monsoon, westerlies, ENSO, NAO, and PDO" it is a general sentence that does not provide critical information for an abstract which is -in my opinion- too long.

Response: Thanks very much for your suggestion. We have removed the sentence.

Lines 35-36: "PMIP3 multi-model simulation between the LGM and Mid-Holocene (MH) in summer and winter visually was…".

The word "visually" is not correctly placed in the sentences. Please rewrite (e.g., "A visual inspection…").

Response: Thanks very much for your suggestion. Based on your suggestion, we have rewritten the sentence to "A visual inspection from PMIP3 multi-model simulations in summer and winter shows that the insolation in different seasons control the intensity of westerlies and summer monsoon and relatively humid and further influence the summer and winter precipitation in EA and CA since the LGM.".

Line 63: "Therefore, exploring spatiotemporal climate and environment changes in EA and CA has attracted much research interest."

In my opinion this sentence does not lay the groundwork for the general question posed in its corresponding paragraph, and looks detached from the previous sentences. I suggest changing by "These contrasting climate regimes have attracted much research interest".

Response: Thanks very much for your suggestion. We have made corresponding revisions according to your suggestion.

Line 67: Could you please provide a definition for "multi-time scales".

Response: Thanks very much for your carefulness. The multi-time scales mentioned here mainly include orbital, millennial, interdecadal and annual timescales, and we have added definitions in the text. We have added the definition in the main text. In addition, the discussion following this sentence provides a detailed review of research on multi-time scales.

Line 75: "…records in part regions of CA.." I suggest changing this text by "…records in some regions of CA".

Response: Thanks very much for your suggestion. Done.

Line 79: I suggest changing "causing" by "caused".

Response: Thanks very much for your suggestion. Done.

Lines 84-88: Please consider rewriting this sentence. It is full of technical jargon with no clear meaning.

Response: Thanks very much for your suggestion. We have rewritten this sentence, mainly splitting it up and removing some meaningless descriptions. The updated sentence is "The seasonal signals of precipitation, derived from the simultaneity of rain and heat periods, is an important climate phenomenon at the multi-time scale. It behaves as that the summer half-year at short-term timescales and warm period at long-term timescales get more precipitation than the winter half-year and cold period respectively.".

Lines 97-100: In this long sentence the authors define the limits of the central Asia region, providing a reference to Figure 1. However, there is no mention where central Asia is in the figure. Particularly useful would be to add to the map where eastern central Asia is located as well. Then again in lines 110-111 the author refers to Figure 1 to mention the central and eastern Asia regions, but no references are in the map.

Response: Thanks very much for your suggestions. We have added a clear depiction

of Eastern Asia, Central Asia, the core region of CA, and the east of CA in Figure 1. Meanwhile, boundaries of CA and EA are added to all images in the study. The updated figure is reproduced below:

"

[Figure]

**Figure. 1** Overview map showing the paleoclimate record sites selected in this study from EA and CA, the difference between summer and winter precipitation over 1965-2014 (shade), and the dominant circulation systems, including the westerlies, Asian winter monsoon and East Asian summer monsoon. The modern Asian summer monsoon limit (red solid line) is summarized by Chen et al. (2008, 2019). The gray slash represents the simultaneous region of the rain and heat periods. The brown solid line represents the range of CA, the core region of CA, the east of CA, and EA as defined in this study.

"

Lines 101-102: monsoon China" Please provide another term for this expression (e.g., The chinese monsoon")

Response: Thanks very much for the comment. According to your suggestion, we amended the description to "we viewed Chinese monsoon region in the east and south of the modern Asian summer monsoon limit in China as EA.".

Line 132: "The high-resolution monthly averaged data high resolution…". Please remove "high resolution" as it is written twice.

Response: Thanks very much for the suggestion. Done.

Lines 149-151: "We used the Hurrell NAO index (station-based) to investigate the impact factor of midlatitude westerlies (Hurrell, 1995; Hurrell and Deser, 2009)". What does "impact factor of midlatitude westerlies" mean? I would suggest adding a short definition of this concept in the main text.

Response: Thanks very much for your suggestion and reminder. Our original intention is to express the influence of NAO on mid-latitude westerlies. Thus, we have amended the sentence to "We used the Hurrell NAO index (station-based) to investigate the impact of NAO on midlatitude westerlies (Hurrell, 1995; Hurrell and Deser, 2009).".

Line 162: Consider changing "index" by "indexes" or "indices".

Response: Thanks very much for your carefulness. Done.

Lines 243-244: "…we conducted the EOF analysis on the seasonal precipitation in spring…". I suggest replacing this sentence with "We conducted EOF analysis on spring precipitation data".

Response: Thanks very much for your suggestion. Done.

Line 391: It would be desirable to add a statement indicating where the Tarim basin is.

Response: Thanks very much for your suggestion. Done.

Line 404: I would suggest replacing "linkage" by "similarities".

Response: Thanks very much for your suggestion. Done.

Lines 448-449: "With recent global warming, some recent work also points out increasing summer precipitation in arid CA". The word "recent" is written twice in the same sentence, please revise.

Response: Thanks very much for your suggestion. We have deleted the first "recent".

**Figures and tables**

Figure 1. It would be great if the first image provided a clear depiction of eastern Asia and central Asia. This will be very useful since there is reference of these regions all throughout the text.

Response: Thanks very much for your suggestion. We have added a clear depiction of Eastern Asia, Central Asia, the core region of CA, and the east of CA in Figure 1. The updated figure is reproduced below:

"

[Figure]

**Figure. 1** Overview map showing the paleoclimate record sites selected in this study from EA and CA, the difference between summer and winter precipitation over 1965-2014 (shade), and the dominant circulation systems, including the westerlies, Asian winter monsoon and East Asian summer monsoon. The modern Asian summer monsoon limit (red solid line) is summarized by Chen et al. (2008, 2019). The gray slash represents the simultaneous region of the rain and heat periods. The brown solid line represents the range of CA, the core region of CA, the east of CA, and EA as defined in this study.

"

Table 1. is in my opinion too long, as it covers 4 pages of the main text. The location of the sites is also denoted in Figure 1. I may suggest adding this long table to the Supplements.

Response: Thanks very much for your suggestion. We have transferred Table 1 to the Supplement materials.

Figure 7. This should be Figure 5

Response: Thanks very much for your carefulness. We have checked the citation of figures throughout the text.

Figure 8. Figure 8 should be labeled "Figure 6".

Response: Thanks very much for your carefulness. We have checked the citation of figures throughout the text.

Figure 5 should be labeled "Figure 7".

Response: Thanks very much for your carefulness. We have checked the citation of figures throughout the text.

---

## Author Comment (AC2)

Dear reviewer,

Thank you for your letter and for the comments concerning our manuscript entitled "Tracing seasonal signals in dry/wet status for regions with simultaneous rain and heat from Eastern and Central Asia since the Last Glacial Maximum". Thanks very much for giving us such a precious chance. Those comments and suggestions are all valuable and very helpful for revising and improving our paper, as well as the important guiding significance to our research. We have studied all of the comments and suggestions carefully and made corresponding corrections in the manuscript with colored text. A detailed list of revisions against each point of reviewer that is being raised is below.

**Reviewer 2**

The authors presented a comprehensive analysis on the short-term and long-term climate changes in the Central Asia and the East Asia by making use of multiple datasets and simulation results. The conclusions are reasonably reliable based on the spatiotemporal correlation results. However, I have to say that most conclusions have been available throughout previous publications. Therefore, the authors are expected to indicate clearly what's new in their works in comparison to previous publications.

Response: We'd like to thank your constructive comments and suggestions on our manuscript. We appreciate the chance to clarify some concepts and have adopted all suggested changes raised by Reviewer 2. In addition, we have scrutinized the text for additional minor errors and to improve clarity and readability, which are also highlighted in the revised version. We sincerely hope this revision meets with your approval. Considering your concern, we would like to reiterate the innovation of the article, hoping to meet your requirements, as follows: First, we traced the impact of seasonal signals in dry/wet status for EA and CA based on the climate phenomenon of simultaneous rain and heat. We divided summer precipitation regimes and winter precipitation regimes based on multi-year summer and winter mean precipitation. A comprehensive study of multi-timescale datasets including modern observations, paleoclimate records, and paleoclimate simulations since the LGM was carried out for

EA and the east of CA with the summer precipitation regimes. Secondly, we clarified the differences and linkages between dry/wet status in EA and CA at multi-time scales. Although there are overall differences in climate change patterns between the two regions, they also share some spatial connections. Finally, we proposed that the seasonal precipitation signal driven by simultaneous rain and heat is an important factor creating this connection between EA and CA. We suggested the common influence of the seasonal signal on EA and the east CA with summer precipitation regimes, both on short-term and long-term timescales. In summary, we analyzed the contradictions in dry/wet status in EA and CA from a new perspective and combined a variety of data to confirm it from various aspects.

1. Generally, definition of the Central Asia and the East Asia is a geographical, or even geopolitical, rather than climatological concept. Misunderstanding may arise certainly when performing spatiotemporal correlation referring to the domains of the Central Asia and the East Asia. For instance, the authors indicated that "In summer precipitation, the centers of positive values are mainly distributed in the north of EA, while the negative values are mainly distributed in CA and south of EA" (Lines 248-250). If this is the case, the north of EA separates from the south of EA, which shares a feature of CA. Climatologically, CA and south of EA should belong to the same domain, while the north of EA belongs to another domain. A similar concern arises out in respect to the winter and annual precipitation.

Response: Thanks very much for your comments. First, we agree with your perspective on defining EA and CA, and indeed, we defined the scope of the study area based on the geographical distribution of EA and CA. Subsequently, we found that by observing the differences in summer and winter precipitation in the EA and CA, non-simultaneous regions of rain and heat periods mostly belong to the core area of CA, while simultaneous regions of rain and heat periods are primarily located in EA and the east of CA (the arid regions of Northwestern China and Southern Mongolia) (China's monsoon region). Now, given your emphasis on the geographical concept, along with previous studies on hydroclimate changes in EA and CA (Chen et al., 2021; Ren et al.,

2021; Qi and Han, 2022; ), we have outlined the region's scope of EA and CA and updated the description in the revised manuscript, as follow:

"

*In this study, we divide the boundaries of CA and EA mainly according to the modern Asian summer monsoon limit designed by Chen et al. (2008, 2019). CA is the largest arid and semi-arid areas in the mid-latitude hinterland of the Eurasian continent, extending from the Caspian Sea in the west to the modern Asian summer monsoon limit in the east, comprising the central Asian countries, NW China, and southern Mongolian Plateau (Fig. 1). Considering that the strength and trajectory of monsoon circulation is a major control on moisture in EA, we viewed Chinese monsoon region in the east and south of the modern Asian summer monsoon limit in China as EA (Fig. 1). We calculated the precipitation difference between the summer (April, May, June, July, August, and September) and winter (January, February, March, October, November, and December) half year over 1971-2020, and then defined the region greater than 0 mm as the simultaneous region of rain and heat periods. Therefore, we define the simultaneous region of rain and heat periods in CA as the east of CA (Fig. 1). The seasonality perspective implies that different precipitation regimes could affect the difference and linkage in climate change modes from EA and CA at the multi-time scale. Taking seasonal signals as the dividing criteria, the core region of CA is characterized by a wet cold-season climate, whereas EA and the east of CA are characterized by a wet warm-season climate (Fig. 1).*

[Figure]

***Figure. 1*** *Overview map showing the paleoclimate record sites selected in this study from EA and CA, the difference between summer and winter precipitation over 1965-2014 (shade), and the dominant circulation systems, including the westerlies, Asian winter monsoon and East Asian summer monsoon. The modern Asian summer monsoon limit (red solid line) is summarized by Chen et al. (2008, 2019). The gray slash represents the simultaneous region of the rain and heat periods. The brown solid line represents the range of CA, the core region of CA, the east of CA, and EA as defined in this study.*

"

Secondly, reanalyzing the EOF of precipitation in the four seasons, we found that the similarities between the summer precipitation in Central EA and the core region of CA are indeed a notable feature. And this feature exists in both summer and winter EOF, but similar EOF results between CA and south of EA do not always exist in winter and summer, therefore, we suggested that this feature may not be able to be specialized. Meanwhile, given the focus of our study on hydroclimate changes in eastern Central Asia, we did not analyze this feature in depth. Based on your comment, we added some new descriptions to avoid misunderstanding.

*Reference:*

*Chen, C et al. (2021): Increasing summer precipitation in arid central Asia linked to the weakening of the East Asian summer monsoon in the recent decades. Int. J. Climatol. 41, 1024-1038.*

*Hu, Q et al. (2022): Northward Expansion of Desert Climate in Central Asia in Recent Decades. Geophys. Res. Lett. 49, e2022GL098895.*

*Ren, Y et al. (2021): Attribution of Dry and Wet Climatic Changes over Central Asia. J. Clim. 35,1399-1421.*

2. The authors studied the seasonal signals at short-term timescales, and seasonal signals at long-term timescales. These two parts are quite distinguished, leaving the contribution is lack of a clear focus. It's certainly true that investigating the past climate is key to informing future climate change, but the authors may need to give an at-least roughly explanation how their works from the past decades and the last glacial period can be applied for such a purpose.

Response: Thanks very much for your suggestions. Based on the analyses of seasonal signals at short-term timescales and long-term timescales, our results provide a hypothesis that seasonal signals of precipitation derived from the simultaneity of rain and heat periods govern the difference and linkage in dry/wet status from EA and CA at multi-time scales. According to your suggestions, we added some speculations about possible future climate scenarios in the final paragraph of our discussion, i.e. "*With global warming and continued increase in winter solar radiation, we suggest that the core region of CA could face a persistent reduction in precipitation in the future. Meanwhile, the decrease in summer solar radiation could lead to a strengthening and southward shift of the summer westerly jet stream over CA, potentially increasing precipitation in the east of CA with summer precipitation regimes. However, more quantitative analyses are required to understand how future interannual variations in atmospheric and oceanic circulation might control the seasonal precipitation signals that influence dry/wet status in the east of CA.*". Moving forward, the focus of our research will be on quantitative analysis of future climate trends, and on comparing them with paleoclimate changes on multi-time scales.

3. The authors presented a compilation of 42 proxy records, which were divided into two groups in name of their precipitation regimes, i.e., winter or summer precipitation

regimes. I missed to find how the authors classificated the 42 proxy records. Especially, Lake Karakul (No. 6 in figure 1) and Lake Issyk- Kul (No.8) are included into winter or summer precipitation regimes, respectively. However, the sediment cores from Lake Karakul and Lake Issyk-Kul shared a common past climate feature (, wetter conditions during the EH and MH periods, see Lines 328-330). If this is the case, what's the reason(s) to label them into different precipitation regime?

Response: Thanks very much for your comments and suggestions. We apologize for this oversight. We defined the winter precipitation regimes and summer precipitation regimes based on the difference between the multi-year average summer and winter precipitation, the results showed that Lake Karakul and Lake Issyk-Kul belong to the winter precipitation regime and the summer precipitation regime, respectively. Figure 5f-i mainly indicates that the paleoclimate record with the summer precipitation regime in the east of CA has a comment dry/wet evolution with the EA mode affected by the East Asian summer monsoon. Therefore, Lake Karakul is not suitable as a typical climate record in this study to compare the dry/wet status in EA and CA and is incorrectly displayed here. Notably, the sediment cores from Lake Karakul with winter precipitation regime definitely have a similar dry/wet status with the paleoclimate records from EA since the LGM. The explanation in the original literature (Heinecke et al., 2017; Aichner et al., 2019) is that wetter conditions during the EH and MH periods are the result of the strengthening of the Indian summer monsoon. For these reasons, we deleted this curve in Figure 5.

Reference:

Aichner, B et al. (2019): Hydroclimate in the Pamirs was driven by changes in precipitation-evaporation seasonality since the last glacial period. Geophysical Research Letters, 46, 13,972–13,983.

Heinecke, L et al. (2017): Climatic and limnological changes at Lake Karakul (Tajikistan) during the last ~29 cal ka. Journal of Paleolimnology, 58(3), 317-334

4. In the abstract, the authors wrote that "seasonal signals of precipitation derived from the simultaneity of rain and heat periods could govern the difference and linkage in dry/wet status from EA and CA. EOF analysis results of mean annual precipitation". I wonder how to get seasonal signals from results of mean annual precipitation.

Response: Thanks very much for your comments. Our description here is not sufficiently clear. The original intent was to indicate that the EOF of mean annual precipitation highlighted the spatial diversity in EA and CA. Subsequently, by analyzing the EOF of precipitation in different seasons, we found similarities between precipitation regimes in EA and the east of CA, further confirming that summer precipitation regimes have a similar impact on EA and the east of CA. Based on your suggestion, we have amended the description in the abstract to "*At short-term timescales, EOF analysis results of mean annual precipitation uncover the spatial diversity of overall precipitation pattern in EA and CA.*".

---

## Author Response (AR2)

Dear reviewer,

Thank you for your letter and for the comments concerning our manuscript entitled "Simultaneous seasonal dry/wet signals in Eastern and Central Asia since the Last Glacial Maximum". Thanks very much for giving us such a precious chance. Those comments and suggestions are all valuable and very helpful for revising and improving our paper, as well as the important guiding significance to our research. We have studied all of the comments and suggestions carefully and made corresponding corrections in the manuscript with colored text. A detailed list of revisions against each point of editor that is being raised is below.

**Reviewer 2**

This version of the title reads easier, but it is your choice.

Response: We agree with this simplification, thank you.

Which monsoon system are you referring to? Please specify.

Response: Thanks very much for your kind suggestion. We have specified the monsoon system.

What is meant here...opposing changes???

Response: Yes. Thanks very much for your kind suggestion.

Acronym needs to be explained. Acronyms not introduced.

Response: Thanks very much for your carefulness. We have added the detailed introduction of acronym.

This does not read well - please reword.

Response: Thanks very much for your suggestion. We have reworded the sentence.

Which oceans - please be specific.

Response: Thanks very much for your suggestion. We have specified the ocean.

Please use dry/wet etc consistently as dry/wet  or dry-wet

Response: Thanks very much for your suggestion. Done.

Please use this phrase instead of "multi-time scales" throughout the manuscript.

Response: Thanks very much for your suggestion. Done.

Be specific - which time scales.

Response: Thanks very much for your suggestion. Done.

It would need a specific geographical region here.

Response: Thanks very much for suggestion. Done.

Reviewed?

Response: This sentence was not expressed clearly enough and we have rewritten it.

Unclear - please reword.

Response: Thanks very much for your suggestion. In fact, this sentence should be deleted. It was an oversight on our part to forget to delete this caption after updating the figure.

Please check the formatting of the S1 table. Also, you would need to say something about comparability of studies, uncertainties of age models and how these data were used in the study.

Response: Thanks very much for your suggestions. We have checked the Table S1 and added detailed introduction for paleoclimate records.

Please check if this acronym has been introduced before. If not, please introduce it here.

Response: Thanks very much for your carefulness. This acronym has been introduced in section 2.4.

Not sure what you want to say here.

Response: Thanks very much for your suggestion. We have amended this description.

At some point in the manuscript, there would need to be a definition of the time slices/windows you have selected from sediment records.

Response: Thangs very much for your suggestion. We have added the detailed definition of four time points (LGM, EH, MH, and LH) in section 2.4.

Reference is missing here.

Response: Thanks very much for your carefulness. Reference has been added.

The sentence was not clear. Not sure if it now reflects what you wanted to say.

Response: Thanks very much for your appropriate modification.

Do you mean to say is controlled by a summer precipitation regime?

Response: Yes, we have modified the sentence based on your suggestion.

Could this be illustrated better. It is not entirely clear from figure 7.

Response: Thanks very much for your suggestion. We have added more detailed illustrations here.

Not sure what is meant here. Is the following what you want to say? "Abundant moisture is brought to CA from polar airmass whereas the North Atlantic and the eastern Mediterranean Sea influence continues to diffuse eastward to the arid region of northwest China ?"

Response: Thanks very much for your suggestion. We have nodified the sentence to make it clear. The updated sentence is "Abundant moisture is carried brought to CA from the polar airmass, North Atlantic and the eastern Mediterranean Sea to CA, and continues to diffuse eastward to reach the arid region of northwest China".

This statement would need to be better explained. Figure 9c on its own does not reveal the complex controls of the summer monsoon strength.

Response: Thanks very much for your suggestion. We have rewritten the sentence to make it clearer.

This should be specified. Figure 8d shows some patchiness. It would therefore be useful to specify which region you are referring to.

Response: Thanks very much for your suggestion. We have added detailed explanation.

See previous comment on patchiness. The same applies to some degree to this statement.

Response: Thanks very much for your suggestion. We have added detailed explanation.

Please add figure reference.

Response: Thanks very much for your carefulness. Done.

This sentence is unclear.

Response: Thanks very much for your suggestion. We have modified the sentence.